# XTC: Extreme Compression for Pre-trained Transformers Made Simple and Efficient

**Xiaoxia Wu**[∗], **Zhewei Yao**[∗], **Minjia Zhang**[∗], **Conglong Li, Yuxiong He**

Microsoft

{xiaoxiawu, zheweiyao, minjiaz, conglong.li, yuxhe}@microsoft.com

## Abstract

Extreme compression, particularly ultra-low bit precision (binary/ternary) quantization, has been proposed to fit large NLP models on resource-constrained devices. However, to preserve the accuracy for such aggressive compression schemes, cutting-edge methods usually introduce complicated compression pipelines, e.g., multi-stage expensive knowledge distillation with extensive hyperparameter tuning. Also, they oftentimes focus less on smaller transformer models that have already been heavily compressed via knowledge distillation and lack a systematic study to show the effectiveness of their methods. In this paper, we perform a very comprehensive systematic study to measure the impact of many key hyperparameters and training strategies from previous works. As a result, we find out that previous baselines for ultra-low bit precision quantization are significantly under-trained. Based on our study, we propose a simple yet effective compression pipeline for extreme compression, named XTC. XTC demonstrates that (1) we can skip the pre-training knowledge distillation to obtain a 5-layer BERT while achieving better performance than previous state-of-the-art methods, e.g., the 6-layer TinyBERT; (2) extreme quantization plus layer reduction is able to reduce the model size by 50x, resulting in new state-of-the-art results on GLUE tasks.

## 1 Introduction

Over the past few years, we have witnessed the model size has grown at an unprecedented speed, from a few hundred million parameters (e.g., BERT [10], RoBERTa [28], DeBERTA [15], T5 [37],GPT-2 [36]) to a few hundreds of billions of parameters (e.g., 175B GPT-3 [6], 530B MT-NLT [46]), showing outstanding results on a wide range of language processing tasks. Despite the remarkable performance in accuracy, there have been huge challenges to deploying these models, especially on resource-constrained edge or embedded devices. Many research efforts have been made to compress these huge transformer models including knowledge distillation [20, 54, 53, 61], pruning [57, 43, 8], and low-rank decomposition [30]. Orthogonally, quantization focuses on replacing the floating-point weights of a pre-trained Transformer network with low-precision representation. This makes quantization particularly appealing when compressing models that have already been optimized in terms of network architecture.

Popular quantization methods include post-training quantization [45, 33, 29], quantization-aware training (QAT) [23, 4, 31, 22], and their variations [44, 24, 12]. The former directly quantizes trained model weights from floating-point values to low precision values using a scalar quantizer, which is simple but can induce a significant drop in accuracy. To address this issue, quantization-aware training directly quantizes a model during training by quantizing all the weights during the forward and using a straight-through estimator (STE) [5] to compute the gradients for the quantizers.

---

[∗]Equal contribution. Code is released as a part of https://github.com/microsoft/DeepSpeed

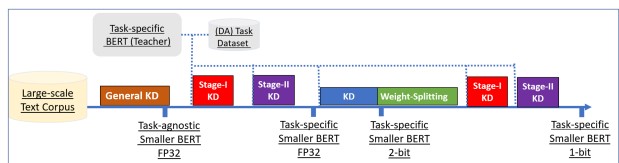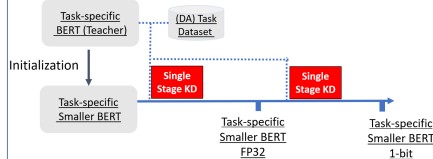

Figure 1: The left figure summarizes how to do 1-bit quantization for a layer-reduced model based on [20, 3]. It involves expensive pretraining on an fp-32 small model, task-specific training on 32-bit and 2-bit models, weight-splitting, and the final 1-bit model training. Along the way, it applies multi-stage knowledge distillation with data augmentation, which needs considerable hyperparameter tuning efforts. The right figure is our proposed method, XTC (see details in § 5), a simple while effective pipeline (see Figure 2 for highlighted results). Better read with a computer screen.

Recently, several QAT works further push the limit of BERT quantization to the extreme via ternarized (2-bit) ([62]) and binarized (1-bit) weights ([3]) together with 4-/8-bit quantized activation. These compression methods have been referred to as **extreme quantization** since the limit of weight quantization, in theory, can bring over an order of magnitude compression rates (e.g., 16-32 times). One particular challenge identified in [3] was that it was highly difficult to perform binarization as there exits a sharp performance drop from ternarized to binarized networks. To address this issue, prior work proposed multiple optimizations where one first trains a ternarized DynaBert [17] and then binarizes with weight splitting. In both phases, multi-stage distillation with multiple learning rates tuning and data augmentation [20] are used. Prior works claim these optimizations are essential for closing the accuracy gap from binary quantization.

While the above methodology is promising, several unanswered questions are related to these recent extreme quantization methods. First, as multiple ad-hoc optimizations are applied at different stages, the compression pipeline becomes very complex and expensive, limiting the applicability of extreme quantization in practice. Moreover, a systematical evaluation and comparison of these optimizations are missing, and the underlying question remains open for extreme quantization:

*what are the necessities of ad-hoc optimizations to recover the accuracy loss?*

Second, prior extreme quantization primarily focused on reducing the precision of the network. Meanwhile, several advancements have also been made in the research direction of knowledge distillation, where large teacher models are used to guide the learning of a small student model. Examples include Distil-BERT [42], MiniLM [54, 53], Mobile-BERT [49], which demonstrate 2-4× model size reduction by reducing the depth or width without much accuracy loss through pre-training distillation with an optional fine-tuning distillation. Notably, TinyBERT [20] proposes to perform deep distillation in both the pre-training and fine-tuning stages and shows that this strategy achieves state-of-the-art results on GLUE tasks. However, most

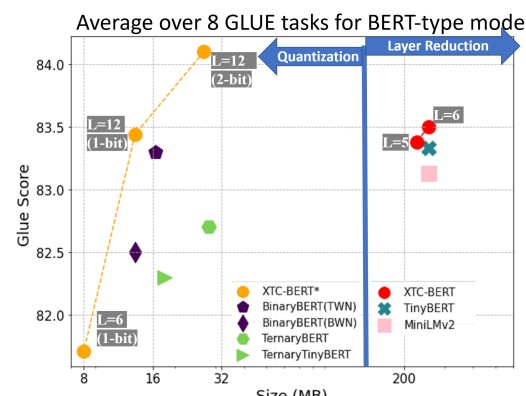

Figure 2: The comparison between XTC with other SOTA results.

of these were done without quantization, and there are few studies about the interplay of extreme quantization with these heavily distilled models, which poses questions on

*to what extent, a smaller distilled model benefits from extreme quantization?*

**Contribution.** To investigate the above questions, we make the following contributions:
(1) We present a systematic study of extreme quantization methods by fine-tuning ≥1000 pre-trained Transformer models, which includes a careful evaluation of the effects of hyperparameters and several

methods introduced in extreme quantization.

(2) We find that previous extreme quantization studies overlooked certain design choices, which lead to under-trained binarized networks and unnecessarily complex optimizations. Instead, we derive a celebrating recipe for extreme quantization, which is not only simpler but also allows us to achieve an even larger compression ratio and higher accuracy than existing methods (see Figure 1, right).

(3) We find that extreme quantization can be effectively combined with lightweight layer reduction, which allows us to achieve greater compression rates for pre-trained Transformers with better accuracy than prior methods while enjoying the additional benefits of flexibly adjusting the size of the student model for each use-case individually, without the expensive pre-training distillation.

Our evaluation results, as illustrated in Figure 2, show that our simple yet effective method can:
(1) compress $BERT_{base}$ to a 5-layer $BERT_{base}$ while achieving better performance than previous state-of-the-art distillation methods, e.g., the 6-layer TinyBERT [20], without incurring the computationally expensive pre-training distillation;
(2) reduce the model size by $50\times$ via performing robust extreme quantization on lightweight layer reduced models while obtaining better accuracy than prior extreme quantization methods, e.g., the 12-layer 1-bit $BERT_{base}$, resulting in new state-of-the-art results on GLUE tasks.

## 2   Related Work

Quantization becomes practically appealing as it not only reduces memory bandwidth consumption but also brings the additional benefit of further accelerating inference on supporting hardware [44, 24, 12]. Most quantization works focus on INT8 quantization [21] or mixed INT8/INT4 quantization [44]. Our work differs from those in that we investigate extreme quantization where the weight values are represented with only 1-bit or 2-bit at most. There are prior works that show the feasibility of using only ternary or even binary weights [62, 3]. Unlike those work, which uses multiple optimizations with complex pipelines, our investigation leads us to introduce a simple yet more efficient method for extreme compression with more excellent compression rates.

On a separate line of research, reducing the number of parameters of deep neural networks models with full precision (no quantization) has been an active research area by applying the powerful knowleadge distillation (KD) [16, 54], where a stronger teacher model guides the learning of another small student model to minimize the discrepancy between the teacher and student outputs. Please see Appendix A for a comprehensive literature review on KD. Instead of proposing a more advanced distillation method, we perform a well-rounded comparative study on the effectiveness of the recently proposed multiple-stage distillation [20].

## 3   Extreme Compression Procedure Analysis

This section presents several studies that have guided the proposed method introduced in Section 5. All these evaluations are performed with the General Language Understanding Evaluation (GLUE) benchmark [51], which is a collection of datasets for evaluating natural language understanding systems. For the subsequent studies, we report results on the development sets after compressing a pre-trained model (e.g., $BERT_{base}$ and TinyBERT) using the corresponding single-task training data.

Previous works [62, 3] on extreme quantization of transformer models state three hypotheses for what can be related to the difficulty of performing extreme quantization:

- Directly training a binarized BERT is complicated due to its irregular loss landscape, so it is better to first to train a ternarized model to initialize the binarized network.

- Specialized distillation that transfers knowledge at different layers (e.g., intermediate layer) and multiple stages is required to improve accuracy.

- Small training data sizes make extreme compression difficult.

### 3.1   Is staged ternary-binary training necessary to mitigate the sharp performance drop?

Previous works demonstrate that binary networks have more irregular loss surface than ternary models using curvature analysis. However, given that rough loss surface is a prevalent issue when training

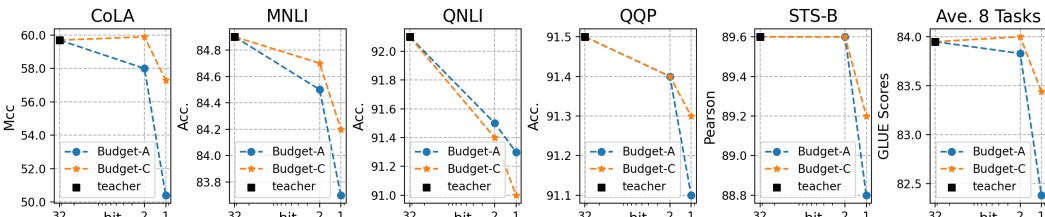

Figure 3: Performance of quantized BERT$_{base}$ with different weight bits and 8-bit activation on the GLUE Benchmarks. The results for orange and blue curves respectively represent the costs: (limited) Budget-A and (sufficient) Budget-C. The fp32-teacher scores are shown by black square marker.

neural networks, we question whether all of them have a causal relationship with the difficulty in extreme compression.

We are first interested in the observation from previous work [3] that shows directly training a binarized network leads to a significant accuracy drop (e.g., up to 3.8%) on GLUE in comparison to the accuracy loss from training a ternary network (e.g., up to 0.6%). They suggest that this is because binary networks have a higher training loss and overall more complex loss surface. To overcome this sharp accuracy drop, they propose a staged ternary-binary training strategy in which one needs first to train a ternarized network and use a technique called weight splitting to use the trained ternary weights to initialize the binary model.

It is not surprising that binarization leads to further accuracy drop as prior studies from computer vision also observe similar phenomenon [18, 50]. However, prior works also identified that the difficulty of training binarized networks mostly resides in training with insufficient number of iterations or having too large learning rates, which can lead to frequent sign changes of the weights that make the learning of binarized networks unstable [50]. Therefore, training a binarized network the same way as training a full precision network consistently achieves poor accuracy. Would increasing the training iterations and let the binarized network train longer under smaller learning rates help mitigate the performance drop from binarization?

To investigate this, we perform the following experiment: We remove the TernaryBERT training stage and weight splitting and directly train a binarized network. We use the same quantization schemes as in [3], e.g., a binary quantizer to quantize the weights: $w_i = \alpha \cdot sign(w_i), \alpha = \frac{1}{n}||w||_1$, and a uniform symmetric INT8 quantizer to quantize the activations. We apply the One-Stage quantization-aware KD that will be explained in § 3.2 to train the model. Nevertheless, without introducing the details of KD here, it does not affect our purpose of understanding the phenomena of a sharp performance drop since we only change the training iteration and learning rates while fixing other setups.

We consider three budgets listed in Table 1, which cover the practical scenarios of short, standard, and long training time, where Budget-A and Budget-B take into account the training details that appeared in BinaryBERT [3], Ternary-BERT [62]. Budget-C has a considerably larger training budget but smaller than TinyBERT [3]. Meanwhile, we also

Table 1: Training budgets for the GLUE tasks.

| Dataset | Data Aug. | Training epochs: Budget-A | Budget-B | Budget-C |
|---|---|---|---|---|
| QQP/MNLI | ✗ | 3 | 9 | 18 or 36 |
| QNLI | ✓ | 1 | 3 | 6 or 9 |
| SST-2/STS-B/RTE | ✓ | 1 | 3 | 12 |
| CoLA/MRPC | ✓ | 1 | 3 | 12 or 18 |

perform a grid search of peak learning rates {2e-5, 1e-4, 5e-4}. For more training details on iterations and batch size per iteration, please see Table C.1.

Table 2: **1-bit** quantization for BERT$_{base}$ with various Budget-A, Budget-B and Budget-C.

| # | Cost | CoLA Mcc | MNLI-m/-mm Acc/Acc | MRPC F1/Acc | QNLI Acc | QQP F1/Acc | RTE Acc | SST-2 Acc | STS-B Pear/Spea | Avg. | Acc Drop |
|---|---|---|---|---|---|---|---|---|---|---|---|
| 0 | Teacher | 59.7 | 84.9/85.6 | 90.6/86.3 | 92.1 | 88.6/91.5 | 72.2 | 93.2 | 90.1/89.6 | 83.95 | - |
| 1 | Budget-A | 50.4 | 83.7/84.6 | 90.0/85.8 | 91.3 | 88.0/91.1 | 72.9 | 92.8 | 88.8/88.4 | 82.38 | 1.57 |
| 2 | Budget-B | 55.6 | 84.1/84.4 | 90.4/86.0 | 90.8 | 88.3/91.3 | 72.6 | 93.1 | 88.9/88.5 | 82.98 | 0.97 |
| 3 | Budget-C | 57.3 | 84.2/84.4 | 90.7/86.5 | 91.0 | 88.3/91.3 | 74.0 | 93.1 | 89.2/88.8 | 83.44 | 0.51 |

We present our main results in Table 2 and Figure 3 (see Table C.2 a full detailed results including three learning rates). We observe that although training binarized BERT with more iterations does not fully close the accuracy gap to the uncompressed full-precision model, the sharp accuracy drop from ternarization to binarization (the blue curve v.s. the orange curve in Figure 3) has largely been mitigated, leaving a much smaller accuracy gap to close. For example, when increasing the training time from Budget-A to Budget-C, the performance of CoLA boosts from 50.4 to 57.3, and the average score improves from 82.38 to 83.44 (Table 2). These results indicate that previous studies [3] on binarized BERT were significantly under-trained. It also suggests that the observed sharp accuracy drop from binarization is primarily an optimization problem. We note that in the context of fine-tuning BERT models, Mosbach et al. [32] also observe that fine-tuning BERT can suffer from vanishing gradients, which causes the training converges to a "bad" valley with sub-optimal training loss. As a result, the authors also suggest increasing the number of iterations to train the BERT model to have stable and robust results.

> **Finding 1.** A longer training iterations with learning rate decay is highly preferred for closing the accuracy gap of extreme quantization.

The above claim seems natural. However, we argue that special considerations need to be taken for effectively doing extreme quantization before resorting to more complex solutions.

## 3.2 The role of multi-stage knowledge distillation

The above analysis shows that the previous binarized BERT models were severely undertrained. In this section, we further investigate the necessity of multi-stage knowledge distillation, which is quite complex because each stage has its own set of hyperparameters. Still, prior works claim multi-stage knowledge distillation to be crucial for improving the accuracy of extreme quantization. Investigating this problem is interesting because it can potentially admit a more straightforward solution with a cheaper cost.

Prior work proposes an interesting knowledge distillation (KD), and here we call it *Two-Stage* KD (2S-KD) [20], which has been applied for extreme quantization [3]. The 2S-KD minimizes the losses of hidden states $\mathcal{L}_{hidden}$ and attention maps $\mathcal{L}_{att}$ and losses of prediction logits $\mathcal{L}_{logit}$ in two separate steps, where different learning rates are used in these two stages, e.g., a $2.5\times$ larger learning rate is used for Stage-I than that for Stage-II. However, how to decide the learning rates and training epochs for these two stages is not well explained and will add additional hyperparameter tuning costs. We note that this strategy is very different from the deep knowledge distillation strategy used in [48, 53, 49], where knowledge distillation is performed with a single stage that minimizes the sum of the losses from prediction logits, hidden states, and attention maps. Despite the promising results in [3], the mechanism for why multi-stage KD improves accuracy is not well understood.

Generally, the knowledge distillation for transformer models can be formulated as minimizing the following objective:

$$\min_{\theta} \mathbb{E}_{x \sim D}[\gamma \mathcal{L}_{logit}(x; \theta) + \beta(\mathcal{L}_{att}(x; \theta) + \mathcal{L}_{hidden}(x; \theta))] \tag{1}$$

where $\mathcal{L}_{logit}$ denote the loss (e.g., KL divergence or mean square error) between student's and teacher's prediction logits, and $\mathcal{L}_{hidden}$ and $\mathcal{L}_{att}$) measures the loss of hidden states and attention maps. $\gamma \in \{0, 1\}, \beta \in \{0, 1\}$ are hyperparameters. See the detailed mathematical definition in § B. To investigate the effectiveness of multi-stage, we compare three configurations (shown in Figure 4 in the appendix):

(1) **1S-KD (One-stage KD):** $(\gamma, \beta) = (1, 1)$ for $t \leq \mathcal{T}$;
(2) **2S-KD (Two-stage KD):** $(\gamma, \beta) = (0, 1)$ if $t < \mathcal{T}/2$; $(\gamma, \beta) = (1, 0)$ if $\mathcal{T}/2 \leq t \leq \mathcal{T}$;
(3) **3S-KD (Three-stage KD):** $(\gamma, \beta) = (0, 1)$ if $t < \mathcal{T}/3$; $(\gamma, \beta) = (1, 1)$ if $\mathcal{T}/3 \leq t \leq 2\mathcal{T}/3$; $(\gamma, \beta) = (1, 0)$ if $2\mathcal{T}/3 \leq t \leq \mathcal{T}$.

where $\mathcal{T}$ presents the total training budget and $t$ represents the training iterations.

Notably, we created the 3S-KD, where we add a transition stage: $\mathcal{L}_{hidden} + \mathcal{L}_{att} + \mathcal{L}_{logit}$ in between the first and second stage of the 2S-KD. The idea is to make the transition of the training objective

smoother in comparison to the 2S-KD. In this case, the learning rate schedules will be correspondingly repeated in three times, and the peak learning rate of Stage-I is ×2.5 higher than that of Stage-II and Stage-III. We apply all three KD methods independently (under a fixed random seed) for binary quantization across different learning rates {2e-5, 1e-4, 5e-4} under the same training budget. We do not tune any other hyperparameters.

Table 3 shows the results under Budget-A (see Table C.2 for Budget-B/C). We note that when the learning rate is fixed to 2e-5 (i.e., Rows 1, 4, and 7), the previously proposed 2S-KD [20] (81.46) does show performance benefits over 1S-KD (80.33). Our newly created 3S-KD (81.57) obtains even better accuracy under the same learning rate. However, multi-stage makes the extreme compression procedure complex and inefficient because it needs to set the peak learning rates differently for different stages. Surprisingly, the simple 1S-KD easily outperforms both 2S-KD and 3S-KD when we slightly increase the search space of the learning rates (e.g., 1e-4 and 5e-4). This means we can largely omit the burden of tuning learning rates and training epochs for each stage in multi-stage by using single-stage but with the same training iteration budget.

> **Finding 2.** Single-stage knowledge distillation with more training budgets and search of learning rates is sufficient to match or even exceed accuracy from multi-stage ones.

Table 3: **1-bit** quantization for BERT$_{base}$ with three different KD under the training Budget-A.

| # | Stages | learning rate | CoLA Mcc | MNLI-m/-mm Acc/Acc | MRPC F1/Acc | QNLI Acc | QQP F1/Acc | RTE Acc | SST-2 Acc | STS-B Pear/Spea | Avg. a row | Avg. the best |
|---|--------|---------------|----------|--------------------|-------------|----------|------------|---------|-----------|-----------------|------------|---------------|
| 1 |           | 2e-5 | 44.6 | 83.1/83.7 | 88.8/83.1 | 91.1 | 87.4/90.7 | 66.1 | **92.8** | 87.8/87.5 | 80.33 |        |
| 2 | One-Stage | 1e-4 | **50.4** | **83.7/84.6** | **90.1/85.3** | **91.3** | **88.0/91.1** | 71.5 | 92.7 | **88.8/88.4** | 82.16 | **82.38** |
| 3 |           | 5e-4 | 42.3 | 83.3/84.1 | 90.0/85.8 | 89.5 | 87.8/90.9 | **72.9** | 92.5 | 88.1/87.8 | 81.04 |        |
| 4 |           | 2e-5 | 48.2 | **83.3/83.8** | 89.3/84.6 | **90.7** | **87.7/90.9** | 70.4 | **92.5** | 88.7/88.4 | 81.46 |        |
| 5 | Two-Stage | 1e-4 | **48.5** | 83.3/83.4 | **90.0/85.5** | 90.4 | 87.4/90.7 | 70.4 | 92.4 | 88.6/88.2 | 81.47 | 81.59 |
| 6 |           | 5e-4 | 16.2 | 74.9/76.4 | 89.7/84.8 | 87.8 | 85.0/89.0 | 68.2 | 91.6 | 86.2/86.5 | 75.01 |        |
| 7 |            | 2e-5 | **49.3** | **83.1/83.6** | 89.5/84.3 | **91.0** | **87.6/90.9** | 70.8 | **92.4** | 88.7/88.3 | 81.57 |        |
| 8 | Three-Stage | 1e-4 | 49.0 | 83.2/83.4 | **89.7/84.8** | 90.9 | 87.6/90.7 | **73.6** | 92.2 | **88.8/88.5** | 81.84 | 81.93 |
| 9 |            | 5e-4 | 29.5 | 81.3/81.9 | 89.1/83.6 | 88.4 | 85.0/89.3 | 66.1 | 91.9 | 84.1/83.9 | 77.34 |        |

### 3.3 The importance of data augmentation

Prior works augment the task-specific datasets using a text-editing technique by randomly replacing words in a sentence with their synonyms, based on their similarity measure on GloVe embeddings [20]. They then use the augmented dataset for task-specific compression of BERT models, observing improved accuracy for extreme quantization [62, 3]. They hypothesized that data augmentation (DA) is vital for compressing transformer models. However, as we have observed that the binarized networks were largely under-trained and had no clear advantage of multi-stage versus single-stage under a larger training budget, it raises the question of the necessity of DA.

To better understand the importance of DA for extreme compression, we compare the end-task performance of both 1-bit quantized BERT$_{base}$ models with and without data augmentation, based on previous findings in this paper that make extreme compression more robust. Table 4 shows results for the comparison of the 1-bit BERT$_{base}$ model under two different training budgets. Note that MNLI /QQP does not use DA; we repeat the results. We find that regardless of whether shorter or longer iterations, removing DA leads to an average of 0.66 and 0.77 points drop in average GLUE score. Notably, the accuracy drops more than 1 point on smaller tasks such as CoLA, MPRC, RTE, STS-B. Furthermore, similar performance degradation is observed when removing DA from training an FP32 half-sized BERT model (more details will be introduced in § 4), regardless of whether using 1S-KD or 2S-KD. These results indicate that DA is helpful, especially for extreme compressed models; DA significantly benefits from learning more diverse/small data to mitigate accuracy drops.

> **Finding 3.** Training without DA hurts performance on downstream tasks for various compression tasks, especially on smaller tasks.

We remark that although our conclusion for DA is consistent with the finding in [3], we have performed a far more well-through experiment than [3] where they consider a single budget and 2S-KD only.

Table 4: **The Comparison between the results with and without data augmentation (DA).** Row 1-4 is for BERT$_{base}$ under 1-bit quantization using 1S-KD. Row 5-8 is for a layer-reduced BERT$_{base}$ (six-layer) under Budget-C without quantization (please see § 4 for more details).

| # | Cost or Stages | Data Aug. | CoLA Mcc | MNLI-m/-mm Acc/Acc | MRPC F1/Acc | QNLI Acc | QQP F1/Acc | RTE Acc | SST-2 Acc | STS-B Pear/Spea | Avg. all | DIFF |
|---|---|---|---|---|---|---|---|---|---|---|---|---|
| 1 | Budget-A | ✓ | 50.4 | 83.7/84.6 | 90.0/85.8 | 91.3 | 88.0/91.1 | 72.9 | 92.8 | 88.8/88.4 | 82.38 | -0.77 |
| 2 | | ✗ | 48.9 | 83.7/84.6 | 89.0/83.6 | 91.0 | 88.0/91.1 | 71.5 | 92.5 | 87.6/87.5 | 81.61 | |
| 3 | Budget-C | ✓ | 57.3 | 84.2/84.4 | 90.7/86.5 | 91.0 | 88.3/91.3 | 74.0 | 93.1 | 89.2/88.8 | 83.44 | -0.66 |
| 4 | | ✗ | 55.0 | 84.2/84.4 | 90.0/85.0 | 90.7 | 88.3/91.3 | 73.3 | 92.9 | 88.2/87.9 | 82.78 | |
| 5 | One-stage | ✓ | 56.9 | 84.5/84.9 | 89.7/84.8 | 91.7 | 88.5/91.5 | 71.8 | 93.5 | 89.8/89.4 | 83.27 | -2.18 |
| 6 | | ✗ | 52.2 | 84.5/84.9 | 88.4/82.4 | 91.4 | 88.5/91.5 | 63.9 | 92.9 | 86.5/86.3 | 81.09 | |
| 7 | Two-stage | ✓ | 56.0 | 84.1/84.2 | 89.7/84.8 | 91.5 | 88.2/91.3 | 71.8 | 93.3 | 89.4/89.1 | 82.93 | -2.03 |
| 8 | | ✗ | 50.2 | 84.1/84.2 | 88.7/83.1 | 90.8 | 88.2/91.3 | 64.6 | 92.9 | 86.9/86.7 | 80.90 | |

# 4 Interplay of KD, Long Training, Data Augmentation and Layer Reduction

In the previous section, we found strong evidence for statistically significant benefits from 1S-KD compared to 2S-KD and 3S-KD. Moreover, a considerable long training can greatly lift the score to the level same as the full precision teacher model. Although 1-bit BERT$_{base}$ already achieves $\times 32$ smaller size than its full-precision counterpart, a separate line of research focuses on changing the model architecture to reduce the model size. Notably, one of the promising methods – reducing model sizes via knowledge distillation – has long been studied and shown good performances [13, 60, 42, 53, 49].

While many works focus on innovating better knowledge distillation (KD) pretraining methods for higher compression ratio and better accuracy, given our observations in Section 3 that extreme quantization with long training and data augmentation can effectively reduce the model size, we are interested in *to what extent can a layer-reduced model benefits from the pretraining with KD?*

To investigate this issue, we perform the following experiments: We prepare five student models, all having 6 layers with the same hidden dimension 768: (1) a pretrained TinyBERT$_6$ [20][2]; (2) MiniLMv2$_6$ [53][3]; (3) Top-BERT$_6$: using the top 6 layers of the fine-tuned BERT$_{base}$ model to initialize the student model; (4) Bottom-BERT$_6$: using the bottom six layers of the fine-tuned BERT$_{base}$ model to initialize the student; (5) Skip-BERT$_6$: using every other layer of the fine-tuned BERT$_{base}$ model to initialize the student. In all cases, we fine-tune BERT$_{base}$ for each task as the teacher model (see the teachers' performance in Table 5, Row 1). We choose TinyBERT and MiniLM because they are the state-of-the-art for knowledge distillation of BERT models. We choose the other three configurations because prior work [48, 34] also suggested that layer pruning is also a practical approach for task-specific compression. For the initialization of the student model, both TinyBERT$_6$ and MiniLM$_6$ are initialized with weights distilled from BERT$_{base}$ through pretraining distillation, and the other three students (e.g., top, bottom, skip) are obtained from the fine-tuned teacher model *without incurring any pretraining training cost.* We apply 1S-KD as we also verify

Table 5: Pre-training does not show benefits for layer reduction. Row 3 (rep.*) is a reproduced result by following the training recipe in [20].

| # | Model | size | CoLA Mcc | MNLI-m/-mm Acc/Acc | MRPC F1/Acc | QNLI Acc | QQP F1/Acc | RTE Acc | SST-2 Acc | STS-B Pear/Spea | Avg. all | Acc. Drop |
|---|---|---|---|---|---|---|---|---|---|---|---|---|
| 1 | fp32 BERT$_{base}$ (teacher) | 417.2 | 59.7 | 84.9/85.6 | 90.6/86.3 | 92.1 | 88.6/91.5 | 72.2 | 93.2 | 90.1/89.6 | 83.95 | - |
| **Training cost: greater than Budget-C (see [20] or § B)** | | | | | | | | | | | | |
| 2 | Pretrained TinyBERT$_6$ ([20]) | 255.2 (×1.6) | 54.0 | 84.5/84.5 | 90.6/86.3 | 91.1 | 88.0/91.1 | 73.4 | 93.0 | 90.1/89.6 | 83.11 | -0.84 |
| 3 | Pretrained TinyBERT$_6$ (rep.*) | 255.2 (×1.6) | 56.9 | 84.4/84.8 | 90.1/85.5 | 91.3 | 88.4/91.4 | 72.2 | 93.2 | 90.3/90.0 | 83.33 | -0.62 |
| **Training cost: Budget-C** | | | | | | | | | | | | |
| 4 | Pretrained TinyBERT$_6$ | 255.2 (×1.6) | 54.4 | 84.6/84.3 | 90.4/86.3 | 91.5 | 88.5/91.5 | 69.7 | 93.3 | 89.2/89.0 | 82.76 | -1.19 |
| 5 | Pretrained MiniLM$_6$-v2 | 255.2 (×1.6) | 55.4 | 84.5/84.5 | 90.7/86.5 | 91.4 | 88.5/91.5 | 71.8 | 93.3 | 89.4/89.0 | 83.13 | -0.82 |
| 6 | Skip-BERT$_6$ (ours) | 255.2 (×1.6) | 56.9 | 84.6/84.9 | 90.4/85.8 | 91.8 | 88.6/91.6 | 72.6 | 93.5 | 89.8/89.4 | **83.50** | -0.45 |
| 7 | Skip-BERT$_5$ (ours) | 228.2 (×1.8) | 57.9 | 84.3/85.1 | 90.1/85.5 | 91.4 | 88.5/91.5 | 72.2 | 93.3 | 89.2/88.9 | **83.38** | -0.57 |
| 8 | Skip-BERT$_4$ (ours) | 201.2 (×2.1) | 53.3 | 83.2/83.4 | 90.0/85.3 | 90.8 | 88.2/91.3 | 70.0 | 93.5 | 88.8/88.4 | 82.18 | -1.77 |

that 2S-KD and 3S-KD are under-performed (See Table C.7 in the appendix). We set Budget-C as our training cost because the training budget in reproducing the results for TinyBERT is greatly larger than Budget-C illustrated above. We report their best validation performances in Table 5 across the three learning rate {5e-5, 1e-4, 5e-4}. We apply 1S-KD and budget-C for this experiment. We

---

[2]The checkpoint of TinyBERT is downloaded from their uploaded huggingface.co.
[3]The checkpoint of miniLMv2 is from their github.

report their best validation performances in Table 5 across three learning rates {5e-5, 1e-4, 5e-4}. For complete statistics with these learning rates, please refer to Table C.5. We make a few observations:

First, perhaps a bit surprisingly, the results in Table 5 show that there are no significant improvements from using the more computationally expensive pre-pretraining distillation in comparison to lightweight layer reduction: Skip-BERT$_6$ (Row 7) achieves the highest average score 83.50. This scheme achieves the highest score on larger and more robust tasks such as MNLI/QQP/QNLI/SST-2.

One noticeable fact in Table 5 is that under the same budget (Budget-C), the accuracy drop of Skip-BERT$_6$ is about 0.74 (0.37) higher than TinyBERT$_6$ in Row 5 (MINILM$_6$-v2 in Row 6). Note that layer reduction without pretraining has also been addressed in [41]. However, their KD is limited to logits without DA, and the performance is not better the pretrained DistillBERT, which shows pretraining was useful (if only logits are used in KD).

Second, with the above encouraging results from a half-size model (Skip-BERT$_6$), we squeeze the depths into five and four layers. The five-/four-layer student is initialized from $\ell$-layer of teacher with $\ell \in \{3, 5, 7, 9, 11\}$ or $\ell \in \{3, 6, 9, 12\}$. We apply the same training recipe as Skip-BERT$_6$ and report the results in Row 8 and 9 in Table 5. There is little performance degradation in our five-layer model (83.28) compared to its six-layer counterpart (83.50); Interestingly, CoLA and MNLI-mm even achieve higher accuracy with smaller model sizes. Meanwhile, when we perform even more aggressive compression by reducing the depth to 4, the result is less positive as the average accuracy drop is around 1.3 points.

Third, among three lightweight layer reduction methods, we confirm that a Skip-# student performs better than those using Top-# and Bottom-#. We report the full results of this comparison in the Appendix Table C.6. This observation is consistent with [20]. However, their layerwise distillation method is not the same as ours, e.g, they use the non-adapted pretrained BERT model to initialize the student, whereas we use the fine-tuned BERT weights. We remark that our finding is in contrast to [41] which is perhaps because the KD in [41] only uses logits distillation without layerwise distillation.

> **Finding 4.** Lightweight layer reduction matches or even exceeds expensive pre-training distillation for task-specific compression.

## 5  Proposed Method for Further Pushing the Limit of Extreme Compression

Based on our studies, we propose a simple yet effective method tailored for extreme compression. Figure 1 (right) or Figure 7 illustrates our proposed method: XTC, which consists of 2 steps:

**Step I: Lightweight layer reduction.** Unlike the common layer reduction method where the layer-reduced model is obtained through computationally expensive pre-training distillation, we select a subset of the fine-tuned teacher weights as a lightweight layer reduction method (e.g., either through simple heuristics as described in Section 4 or search-based algorithm as described in [34]) to initialize the layer-reduced model. When together with the other training strategies identified in this paper, we find that such a lightweight scheme allows for achieving a much larger compression ratio while setting a new state-of-the-art result compared to other existing methods.

**Step II: 1-bit quantization by applying 1S-KD with DA and long training.** Once we obtain the layer-reduced model, we apply the quantize-aware 1S-KD, proven to be the most effective in § 3.2. To be concrete, we use an ultra-low bit (1-bit/2-bit) quantizer to compress the layer-reduced model weights for a forward pass and then use STE during the backward pass for passing gradients. Meanwhile, we minimize the single-stage deep knowledge distillation objective with data augmentation enabled and longer training Budget-C (such that the training loss is close to zero).

**Evaluation Results.** The results of our approach are presented in Table 6, which we include multiple layer-reduced models with both 1-bit and 2-bit quantization. We make the following observations: **(1)** For the 2-bit + 6L model (Row 1 and Row 2), XTC achieves 0.63 points higher accuracy than 2-bit quantized model TernaryBERT [62]. This remarkable improvement consists of two factors: (a) our fp-32 layer-reduced model is better, and (b) the 1-bit model is trained under larger budget and three learning rate searches. To understand how much the factor-(b) benefits, we may refer to the accuracy drop from their fp32 counterparts (last col.) where ours only drops 0.44 points and

Table 6: 1-/2-bit quantization of the layer-reduced model. The last column (Acc. drop) is the accuracy drop from their own fp-32 models. See full details in Table C.8.

| # | Method | bit (#-layer) | size (MB) | CoLA Mcc | MNLI-m/-mm Acc/Acc | MRPC F1/Acc | QNLI Acc | QQP F1/Acc | RTE Acc | SST-2 Acc | STS-B Pear/Spea | Avg. all | Acc. drop |
|---|--------|---------------|-----------|----------|---------------------|-------------|----------|------------|---------|-----------|------------------|----------|-----------|
| 1 | [62] | 2 (6L) | 16.0 (×26.2) | 53.0 | 83.4/83.8 | 91.5/88.0 | 89.9 | 87.2/90.5 | 71.8 | 93.0 | 86.9/86.5 | 82.26 | -0.76 |
| 2 | | 2 (6L) | 16.0 (×26.2) | 53.8 | 83.6/84.2 | 90.5/86.3 | 90.6 | 88.2/91.3 | 73.6 | 93.6 | 89.0/88.7 | **82.89** | -0.44 |
| 3 | | 2 (5L) | 14.2 (×29.3) | 53.9 | 83.3/84.1 | 90.4/86.0 | 90.4 | 88.2/91.2 | 71.8 | 93.0 | 88.4/88.0 | **82.46** | -0.72 |
| 4 | Ours | 2 (4L) | 12.6 (×33.2) | 50.3 | 82.5/83.0 | 90.0/85.3 | 89.2 | 87.8/91.0 | 69.0 | 92.8 | 87.9/87.4 | 81.22 | -0.90 |
| 5 | | 1 (6L) | 8.0 (×52.3) | 52.3 | 83.4/83.8 | 90.0/85.3 | 89.4 | 87.9/91.1 | 68.6 | 93.1 | 88.4/88.0 | 81.71 | -1.51 |
| 6 | | 1 (5L) | 7.1 (×58.5) | 52.2 | 82.9/83.2 | 89.9/85.0 | 88.5 | 87.6/90.8 | 69.3 | 92.9 | 87.3/87.0 | 81.34 | -1.84 |
| 7 | | 1 (4L) | 6.3 (×66.4) | 48.3 | 82.0/82.3 | 89.9/85.5 | 87.7 | 86.9/90.4 | 63.9 | 92.4 | 87.1/86.7 | 79.96 | -2.01 |

TinyBERT$_6$ drops 0.76. **(2)** When checking between our 2-bit five-layer (Row 3, 82.46) in Table 6 and 2-bit TinyBERT$_6$ (Row 1, 82.26), our quantization is much better, while our model size is 12.5% smaller, setting a new state-of-the-art result for 2-bit quantization with this size. **(3)** Let us take a further look at accuracy drops (last column) for six/five/four layers, which are 0.44 (1.51), 0.72 (1.84), and 0.9 (2.01) respectively. It is clear that smaller models become much more brittle for extreme quantization than BERT$_{base}$, especially the 1-bit compression as the degradation is ×3 or ×4 higher than the 2-bit quantization.

Besides the reported results above, we have also done a well-thorough investigation on the effectiveness of adding a low-rank (LoRa) full-precision weight matrices to see if it will further push the accuracy for the 1-bit layer-reduced models. However, we found a negative conclusion on using LoRa. Interestingly, if we repeat another training with the 1-bit quantization after the single-stage long training, there is another 0.3 increase in the average GLUE score (see Table C.14). Finally, we also verify that prominent teachers can indeed help to improve the results (see appendix).

## 6 Conclusions

We carefully design and perform extensive experiments to investigate the contemporary existing extreme quantization methods [20, 3] by fine-tuning pre-trained BERT$_{base}$ models with various training budgets and learning rate search. Unlike [3], we find that there is no sharp accuracy drop if long training with data augmentation is used and that multi-stage KD and pretraining introduced in [20] is not a must in our setup. Based on the finding, we derive a user-friendly celebrating recipe for extreme quantization (see Figure 1, right), which allows us to achieve a larger compression ratio and higher accuracy. See our summarized results in Table 7.[4]

**Discussion and future work.** Despite our encouraging results, we note that there is a caveat in our **Step I** when naively applying our method (i.e., initializing the student model as a subset of the teacher) to reduce the width of the model as the weight matrices dimension of teacher do not fit into that of students. We found that the accuracy drop is considerably sharp. Thus, in this domain, perhaps pertaining distillation [20] could be indeed useful, or Neural Architecture Search (AutoTinyBERT [58]) could be another promising direction. We acknowledge that, in order to make a fair comparison with [62, 3], our investigations are based on the classical 1-bit [39] and 2-bit [27] algorithms and without quantization on layernorm [2].

Table 7: Summary of task-specific performance of MNLI and GLUE scores. Also see Figure 2.

| # | Model (8-bit activation) | size (MB) | MNLI-m/-mm | GLUE Score |
|---|--------------------------|-----------|------------|------------|
| 1 | BERT$_{base}$ (Teacher) | 417.2 (×1.0) | 84.9/85.6 | 83.95 |
| 2 | TinyBERT$_6$ (rep*) | 255.2 (×1.6) | 84.4/84.8 | 83.33 |
| 3 | XTC-BERT$_6$ (ours) | 255.2 (×1.6) | **84.6**/84.9 | **83.50** |
| 4 | XTC-BERT$_5$ (ours) | 228.2 (×1.8) | 84.3/**85.1** | 83.38 |
| 5 | 2-bit BERT [62] | 26.8 (×16.0) | 83.3/83.3 | 82.73 |
| 6 | 2-bit XTC-BERT (ours) | 26.8 (×16.0) | **84.6/84.7** | **84.10** |
| 7 | 1-bit BERT (TWN) [3] | 16.5 (×25.3) | 84.2/**84.7** | 82.49 |
| 8 | 1-bit BERT (BWN) [3] | 13.4 (×32.0) | 84.2/84.0 | 82.29 |
| 9 | 1-bit XTC-BERT (ours) | 13.4 (×32.0) | 84.2/84.4 | **83.44** |
| 10 | 2-bit TinyBERT$_6$ [62] | 16.0 (×26.2) | 83.4/83.8 | 82.26 |
| 11 | 2-bit XTC-BERT$_6$ (ours) | 16.0 (×26.2) | **83.6/84.2** | **82.89** |
| 12 | 2-bit XTC-BERT$_5$ (ours) | 14.2 (×29.3) | 83.3/84.1 | 82.46 |
| 13 | 1-bit XTC-BERT$_6$ (ours) | 8.0 (×52.3) | 83.4/83.8 | 81.71 |
| 14 | 1-bit XTC-BERT$_5$ (ours) | 7.1 (×58.5) | 82.9/83.2 | 81.34 |

Exploring the benefits of long training on an augmented dataset with different 1-/2-bit algorithms (with or without quantization on layer norm) would be of potential interests [24, 35]. Finally, as our experiments focus on BERT$_{base}$ model, future work can be understanding how our conclusion transfers to decoder models such as GPT-2/3 [36, 6].

---

[4]We decide not to include many other great works [26, 59, 42, 35] since they do not use data augmentation or their setups are not close.

## Acknowledgments

This work is done within the DeepSpeed team in Microsoft. We appreciate the helps from the DeepSpeed team. Particularly, we thank Jeff Rasley for coming up the name of our method and Elton Zheng for solving the engineering issue. We thank the engineering supports from the Turing team in Microsoft.

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
