# A    Additional Related Work

KD has been extensively applied to computer vision and NLP tasks [52] since its debut. On the NLP side, several variants of KD have been proposed to compress BERT [10], including how to define the knowledge that is supposed to be transferred from the teacher BERT model to the student variations. Examples of such knowledge definitions include output logits (e.g., DistilBERT [42]) and intermediate knowledge such as feature maps [48, 1, 63] and self-attention maps [54, 49] (we refer KD using these additional knowledge as deep knowledge distillation [54]). To mitigate the accuracy gap from reduced student model size, existing work has also explored applying knowledge distillation in the more expensive pre-training stage, which aims to provide a better initialization to the student for adapting to downstream tasks. As an example, MiniLM [54] and MobileBERT [49] advance the state-of-the-art by applying deep knowledge distillation and architecture change to pre-train a student model on the general-domain corpus, which then can be directly fine-tuned on downstream tasks with good accuracy. TinyBERT [20] proposes to perform deep distillation in both the pre-training and fine-tuning stage and shows that these two stages of knowledge distillation are complementary to each other and can be combined to achieve state-of-the-art results on GLUE tasks. Our work differs from these methods in that we show how lightweight layer reduction without expensive pre-training distillation can be effectively combined with extreme quantization to achieve another order of magnitude reduction in model sizes for pre-trained Transformers.

# B    Additional Details on Methodology, Experimental Setup and Results

## B.1    Knowledge Distillation

Knowledge Distillation (KD) [16] has been playing the most significant role in overcoming the performance degradation of model compression as the smaller models (i.e., student models) can absorb the rich knowledge of those uncompressed ones (i.e., teacher models) [40, 25, 43, 14]. In general, KD can be expressed as minimizing the difference ($\mathcal{L}_{kd}$) between outputs of a teacher model $T$ and a student model $S$. As we are here studying transformer-based model and that the student and teacher models admit the same number (denoted as $h$) of attention heads, our setup focuses on a particular widely-used paradigm, layerwise KD [20, 62, 3]. That is, *for each layer* of a student model $S$, the loss in the knowledge transferred from teacher $T$ consist of three parts:

(1) the fully-connected hidden states: $\mathcal{L}_{\text{hidden}} = \mathbf{MSE}(\mathbf{H}^S, \mathbf{H}^T)$, where $\mathbf{H}^S, \mathbf{H}^T \in \mathbb{R}^{l \times d_h}$;

(2) the attention maps: $\mathcal{L}_{\text{att}} = \frac{1}{h} \sum_{i=1}^h \mathbf{MSE}(\mathbf{A}_i^S, \mathbf{A}_i^T)$, where $\mathbf{A}_i^S, \mathbf{A}_i^T \in \mathbb{R}^{l \times l}$;

(3) the prediction logits: $\mathcal{L}_{\text{logit}} = \mathbf{CE}(\mathbf{p}^S, \mathbf{p}^T)$ where $\mathbf{p}^S, \mathbf{p}^T \in \mathbb{R}^c$;

where $\mathbf{MSE}$ stands for the mean square error and $\mathbf{CE}$ is a cross entropy loss. Notably, for the first part in the above formulation, $\mathbf{H}^S$ ($\mathbf{H}^T$) corresponds to the output matrix of the student's (teacher's) hidden states; $l$ is the sequence length of the input (in our experiments, it is set to be 64 or 128); $d_h$ is the hidden dimension (768 for BERT$_{base}$). For the second part $\mathbf{A}_i^S$ ($\mathbf{A}_i^T$) is the attention matrix corresponds to the $i$-th heads (in our setting, $h = 12$). In the final part, the dimension $c$ in logit outputs ($\mathbf{p}^S$ and $\mathbf{p}^T$) is either to be 2 or 3 for GLUE tasks. For more details on how the weight matrices involved in the output matrices, please see [20].

We have defined the three types of KD: 1S-KD, 2S-KD, and 3S-KD in main text. See Figure 4 for a visualization. Here we explain in more details. **One-Stage** KD means we naively minimize the sum of teacher-student differences on hidden-states, attentions and logits. In this setup, a single one-time learning rate schedule is used; in our case, it is a linear-decay schedule with warm-up steps 10% of the total training time. **Two-Stage** KD first minimizes the losses of hidden-states $\mathcal{L}_{\text{hidden}}$ and attentions $\mathcal{L}_{\text{att}}$, then followed by the loss of logits $\mathcal{L}_{\text{logit}}$. This type of KD is proposed in [20] and has been used in [3] for extreme quantization. During the training, the (linear-decay) learning rate schedule will be repeated from the first stage to the second one; particularly, in [20] they used a $\times 2.5$ peak learning rate for Stage-I than that for Stage-II. Finally, **Three-Stage** KD succeeds the properties of 1S-KD and 2S-KD. That is, after minimizing the losses of hidden-states $\mathcal{L}_{\text{hidden}}$ and attentions $\mathcal{L}_{\text{att}}$, we add a transition phase: $\mathcal{L}_{\text{hidden}} + \mathcal{L}_{\text{att}} + \mathcal{L}_{\text{logit}}$, instead of directly optimizing $\mathcal{L}_{\text{logit}}$. The learning rate schedules will be correspondingly repeated in three times and the peak learning rate of Stage-I is $\times 2.5$ higher than that of Stage-II and Stage-III.

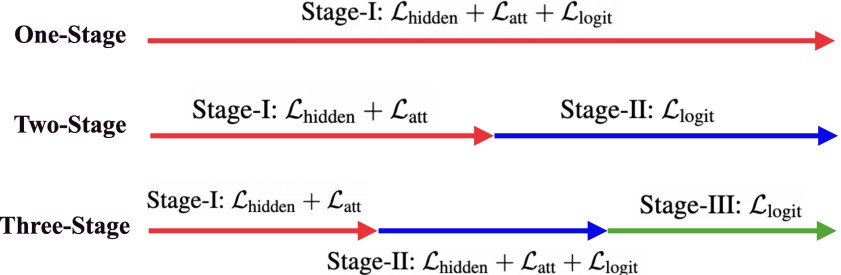

Figure 4: **Three types of knowledge distillation.** 1S-KD KD (top red arrow line) involves all the outputs of hidden-states, attentions and logits from the beginning of the training to the end. 2S-KD KD (middle red and blue arrow line) separates hidden-states and attentions from the logits part. While 3S-KD KD (bottom red, blue and green arrow line) succeed 2S-KD one, it also adds a transition phase in the middle of the training.

## B.2 Experimental Setup

Similar to [20, 3], we fine-tune on GLUE tasks, namely, MRPC [11], STS-B [7], SST-2 [47], QNLI [38], QQP [19], MNLI [56], CoLA [55], RTE [9]). We recorded their validation performances over the training and report the best validation value. The maximum sequence length is set to 64 for CoLA/SST-2, and 128 for the rest sequence pair tasks. See Table C.1 for the size of each (augmented) dataset and training batch size. Each independent experiment is ran on a single A100 GPU with a fixed random seed 42. To understand how much the variance can bring with various random seed, we include a set of experiments by only varying the random seeds 111 and 222. The results are given in Table C.16 and Table C.17. We overall find the standard deviation are within 0.1 on the GLUE score. Note also that we add extra 1 epoch for the warmup training.

The 2-bit algorithm and distillation code are based on the official implementation of [62] [5] and the 1-bit algorithm is based only on the binary code[6] in [3] as explained in the main text. Finally, we comment on the reproduced result shown in Row 3 of Table 5. We apply the 2S-KD where Stage-I distillation requires much longer epochs than Budget-C as ColA requires 50 epochs, MNLI/QQP/QNLI requires 10 epochs, and others 20 epochs. All are trained on augmented datasets; Note that 10-epoch augmented MNLI/QQP is much longer than 18/36-epoch standard ones. Stage-II KD is similar to Budget B but needs to search with three learning rates and two batch sizes.

## B.3 Exploration of LoRa

As we see in § 5 (i.e.,Table 6), smaller models is much more brittle on extreme quantization than $BERT_{base}$, especially the 1-bit compression. We now investigate whether adding low-rank (LoRa) matrices to the quantize weights would help improve the situation. That is, we introduce two low-rank matrices $U \in \mathbb{R}^{d_{in} \times r}$ and $V \in \mathbb{R}^{r \times d_{out}}$ for a quantized weight $\widetilde{W} \in \mathbb{R}^{d_{in} \times d_{out}}$ such that the weight we used for input is

$$W = \widetilde{W} + UV \tag{2}$$

Here $U$ and $V$ are computed in full precision. In our experiments for the layer-reduced model (Skip-Bert$_6$), we let $r \in \{1, 8\}$. We use 1S-KD and also include the scenarios for the three budgets and three learning rates {5e-5,1e-4,5e-4}. The full results for 1-bit and 2-bit Skip-Bert$_6$ are presented in Table C.11 and Table C.12. Correspondingly, we summarized the results in Table C.9 and Table C.10 for clarity.

Overall, we see that LoRa can boost performance when the training time is short. For instance, as shown in Table C.9 ( Table C.10) under Budget-A, the average score for 1-bit (2-bit) is 80.81 (82.14) without LoRa, compared with 81.09 (82.31) with LoRa. However, the benefit becomes marginal when training is long. Here, under Budget-B, the average score for 1-bit (2-bit) is 81.37 (82.51) without LoRa, compared with 81.59 (82.41) with LoRa.

---

[5]https://github.com/huawei-noah/Pretrained-Language-Model/tree/master/TernaryBERT
[6]https://github.com/huawei-noah/Pretrained-Language-Model/blob/master/BinaryBERT/transformer/utils_quant.py#L347

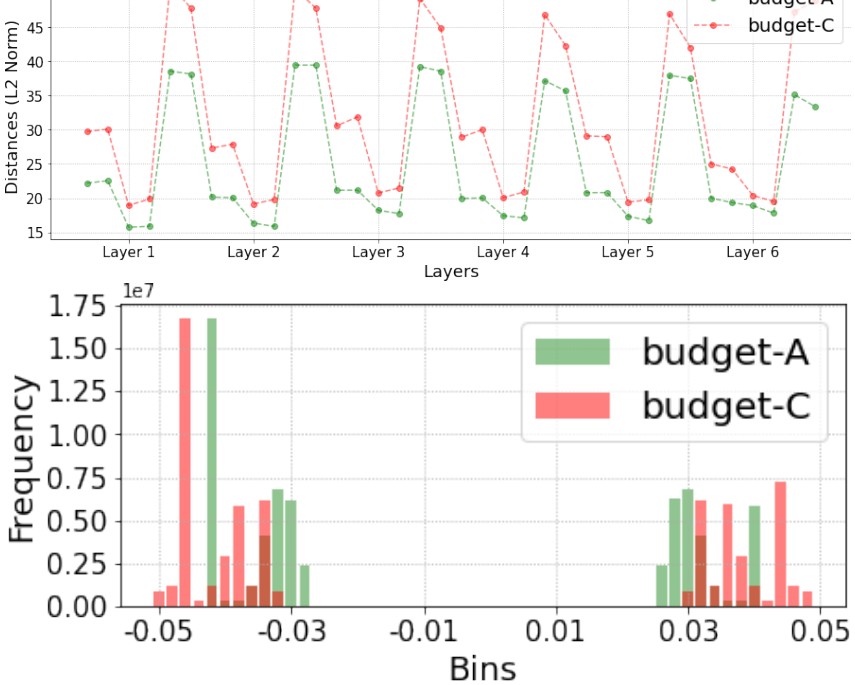

Figure 5: **Top**: The $\ell_2$ norm distance between the fp32 SkipBERT$_6$ and its 1-bit counterpart under Budget-A (green) and Budget-C (red) training recipes (quantizing the fine-tuned 6-layer SkipBERT$_6$ on the RTE task with learning rate 2e-5 and 1S-KD). In each layer, we have 6 linear modules (so there are six points in each layer). **Bottom**: Histograms of the binary weight parameters with Budget-A (green) and Budget-C (red) training recipes (models are the same as the top figure). In addition to the $6 \times 6$ matrices, the elements of binary weights also include the quantized word embedding matrix.

To further verify the claim of "marginal or none" benefit using LoRa under a sufficient training budget, we include more experiments for five-/four-layer BERT (Skip-BERT$_5$ and Skip-BERT$_4$), and the results are shown in Table C.13. The observation is similar to what we have found for Skip-BERT$_6$.

Interestingly, we find that in Table C.14, instead of adding these full-precision low-rank matrices, there is a considerable improvement by continuing training with another Budget-C iteration based on the checkpoints obtained under Budget-C. Particularly, there is 0.09 in GLUE score's improvement for 1-bit SkipBERT$_5$ (from 81.34 (Row 1) to 81.63 (Row 2)) and 0.28 for 1-bit SkipBERT$_4$. On the other hand, if we trained a 2-bit SkipBERT$_5$/SkipBERT$_4$ first and then continue training them into 1-bit, the final performance (see Row 5 and Row 11) is even worse than a direct 1-bit quantization (see Row 1 and Row 7). In addition to the results above, we also tried to see if continuing training by adding LoRa (rank=1) matrices would help or not. However, the results are negative.

### B.4 Exploration of Smaller Teacher

As shown in previous works, such as [53, 25], there is a clear advantage to using larger teacher models. Here shown in Table C.15 under Budget-C, we apply 1-/2-bit quantization on Skip-BERT$_6$ and compare the teachers between BERT$_{base}$ and the full-precision Skip-BERT$_6$.

We verify that a better teacher can boost the performance, particularly for 1-bit quantization on the smaller tasks such as CoLA, MRPC, RTE, and STS-B. However, when adding LoRa, the advantage of using better teacher diminishes as the gain on average GLUE score decreases from 0.31 (without LoRa) to 0.11 (with LoRa) on 1-/2-bit quantization.

## C   Figures and Tables

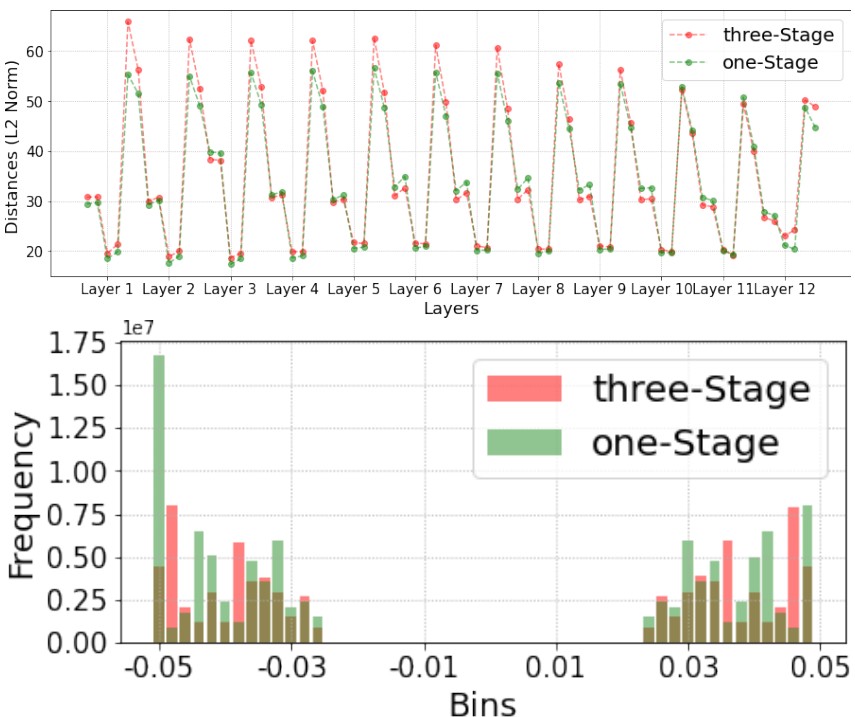

Figure 6: **Top**: The $\ell_2$ norm distance between the fp32 $BERT_{base}$ and its 1-bit counterpart under 3S-KD (red) and 1S-KD (green) training recipes (quantizing the fine-tuned 12-layer $BERT_{base}$ on the RTE task with learning rate 5e-4 and Budget-B). In each layer, we have 6 linear modules (so there are six points in each layer). **Bottom**: Histograms of the binary weight parameters with 3S-KD (red) and 1S-KD (green) training recipes (models are the same as the top figure). In addition to the $12 \times 6$ matrices, the elements of binary weights also include the quantized word embedding matrix.

Table C.1: Training details. DA is short for data augmentation.

|  | CoLA | MNLI | MRPC | QNLI | QQP | RTE | SST-2 | STS-B |
|---|---|---|---|---|---|---|---|---|
| Dev data | 1043 | 9815/9832 | 408 | 5463 | 40430 | 277 | 872 | 1500 |
| Train (noDA) | 8551 | 392702 | 3668 | 104743 | 363846 | 2490 | 67349 | 5749 |
| Train (DA) | 218844 | 392702 | 226123 | 4273773 | 363846 | 145865 | 1133834 | 327384 |
| DA/noDA | 25.6 | 1 | 61.6 | 40.8 | 1 | 58.6 | 16.8 | 56.9 |
| Train batch-size | 16 | 32 | 32 | 32 | 32 | 32 | 32 | 32 |

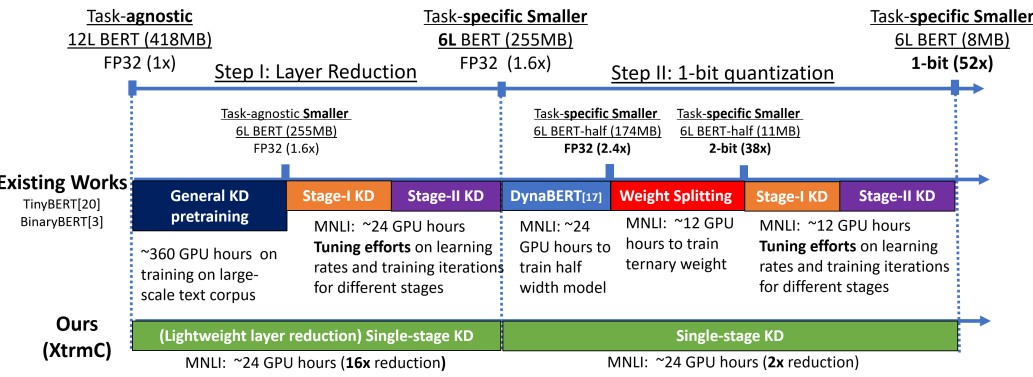

Figure 7: An example on the GPU hours comparing between our methods and the existing works.

Table C.2: 1-bit quantization for BERT$_{base}$ with size 13.0MB ($\times 32.0$ smaller than the fp32 version).

| Training Cost | Stages | learning rate | CoLA Mcc | MNLI-m/-mm Acc/Acc | MRPC F1/Acc | QNLI Acc | QQP F1/Acc | RTE Acc | SST-2 Acc | STS-B Pear/Spea | Avg. | Avg. w/o CoLA |
|---|---|---|---|---|---|---|---|---|---|---|---|---|
| Budget-A | One-Stage | 2e-5 | 44.6 | 83.1/83.7 | 88.8/83.1 | 91.1 | 87.4/90.7 | 66.1 | 92.8 | 87.8/87.5 | 80.33 | 84.80 |
| | | 1e-4 | 50.4 | 83.7/84.6 | 90.1/85.3 | 91.3 | 88.0/91.1 | 71.5 | 92.7 | 88.8/88.4 | 82.16 | 86.12 |
| | | 5e-4 | 42.3 | 83.3/84.1 | 90.0/85.8 | 89.5 | 87.8/90.9 | 72.9 | 92.5 | 88.1/87.8 | 81.04 | 85.89 |
| | | Best (above) | 50.4 | 83.7/84.6 | 90.0/85.8 | 91.3 | 88.0/91.1 | 72.9 | 92.8 | 88.8/88.4 | 82.38 | 86.37 |
| | Two-Stage | 2e-5 | 48.2 | 83.3/83.8 | 89.3/84.6 | 90.7 | 87.7/90.9 | 70.4 | 92.5 | 88.7/88.4 | 81.46 | 85.61 |
| | | 1e-4 | 48.5 | 83.3/83.4 | 90.0/85.5 | 90.4 | 87.4/90.7 | 70.4 | 92.4 | 88.6/88.2 | 81.47 | 85.59 |
| | | 5e-4 | 16.2 | 74.9/76.4 | 89.7/84.8 | 87.8 | 85.0/89.0 | 68.2 | 91.6 | 86.2/86.5 | 75.01 | 82.36 |
| | | Best (above) | 48.5 | 83.3/83.8 | 90.0/85.5 | 90.7 | 87.7/90.9 | 70.4 | 92.5 | 88.7/88.4 | 81.59 | 85.72 |
| | Three-Stage | 2e-5 | 49.3 | 83.1/83.6 | 89.5/84.3 | 91.0 | 87.6/90.9 | 70.8 | 92.4 | 88.7/88.3 | 81.57 | 85.60 |
| | | 1e-4 | 49.0 | 83.2/83.4 | 89.7/84.8 | 90.9 | 87.6/90.7 | 73.6 | 92.2 | 88.8/88.5 | 81.84 | 85.95 |
| | | 5e-4 | 29.5 | 81.3/81.9 | 89.1/83.6 | 88.4 | 85.0/89.3 | 66.1 | 91.9 | 84.1/83.9 | 77.34 | 83.32 |
| | | Best (above) | 49.3 | 83.2/83.4 | 89.7/84.8 | 91.0 | 87.6/90.9 | 73.6 | 92.4 | 88.8/88.5 | 81.93 | 86.01 |
| Budget-B | One-Stage | 2e-5 | 51.8 | 84.1/84.4 | 89.6/84.6 | 90.8 | 88.0/91.1 | 70.8 | 92.8 | 88.8/88.4 | 82.13 | 85.92 |
| | | 1e-4 | 55.6 | 83.9/84.6 | 90.4/86.0 | 90.8 | 88.3/91.3 | 72.6 | 93.1 | 88.9/88.5 | 82.98 | 86.40 |
| | | 5e-4 | 45.7 | 82.9/83.4 | 90.4/86.0 | 89.2 | 87.7/90.8 | 72.6 | 92.8 | 87.8/87.5 | 81.24 | 85.69 |
| | | Best (above) | 55.6 | 84.1/84.4 | 90.4/86.0 | 90.8 | 88.3/91.3 | 72.6 | 93.1 | 88.9/88.5 | 82.98 | 86.40 |
| | Two-Stage | 2e-5 | 52.4 | 84.0/84.5 | 89.7/85.0 | 90.5 | 88.0/91.1 | 72.2 | 92.7 | 89.2/88.8 | 82.40 | 86.15 |
| | | 1e-4 | 51.3 | 83.3/83.6 | 90.3/85.8 | 90.2 | 87.0/90.6 | 72.9 | 92.5 | 88.6/88.3 | 82.09 | 85.94 |
| | | 5e-4 | 29.7 | 78.4/79.6 | 89.2/84.3 | 88.8 | 83.2/88.2 | 69.7 | 91.3 | 84.8/85.1 | 77.20 | 83.14 |
| | | Best (above) | 52.4 | 84.0/84.5 | 90.3/85.8 | 90.5 | 88.0/91.1 | 72.9 | 92.7 | 89.2/88.8 | 82.57 | 86.34 |
| | Three-Stage | 2e-5 | 53.0 | 83.8/84.4 | 90.1/85.3 | 90.7 | 88.0/91.1 | 72.2 | 93.2 | 89.2/88.8 | 82.54 | 86.24 |
| | | 1e-4 | 51.5 | 83.4/83.9 | 90.4/86.0 | 90.2 | 87.9/91.1 | 72.6 | 92.9 | 88.7/88.4 | 82.26 | 86.10 |
| | | 5e-4 | 35.6 | 80.2/80.9 | 89.1/84.6 | 88.5 | 86.1/89.7 | 69.3 | 90.6 | 86.4/86.2 | 78.42 | 83.78 |
| | | Best (above) | 53.0 | 83.8/84.4 | 90.4/86.0 | 90.7 | 88.0/91.1 | 72.6 | 93.2 | 89.2/88.8 | 82.67 | 86.38 |
| Budget-C | One-Stage | 2e-5 | 57.3 | 84.1/84.0 | 90.7/86.5 | 91.0 | 88.2/91.2 | 74.0 | 93.1 | 89.2/88.8 | 83.38 | 86.64 |
| | | 1e-4 | 55.6 | 84.2/84.4 | 90.4/86.0 | 90.4 | 88.3/91.3 | 73.6 | 93.0 | 88.8/88.4 | 83.03 | 86.46 |
| | | 5e-4 | 46.1 | 82.8/82.9 | 89.9/86.0 | 89.5 | 87.7/90.8 | 71.1 | 92.4 | 88.2/88.0 | 81.09 | 85.46 |
| | | Best (above) | 57.3 | 84.2/84.4 | 90.7/86.5 | 91.0 | 88.3/91.3 | 74.0 | 93.1 | 89.2/88.8 | 83.44 | 86.71 |
| | Two-Stage | 2e-5 | 55.9 | 84.2/83.9 | 90.6/86.5 | 90.7 | 88.3/91.3 | 73.3 | 93.1 | 89.2/88.9 | 83.12 | 86.52 |
| | | 1e-4 | 52.0 | 83.2/83.8 | 90.9/86.5 | 90.1 | 88.0/91.0 | 74.0 | 93.1 | 87.8/87.8 | 82.39 | 86.19 |
| | | 5e-4 | 32.2 | 79.3/80.3 | 90.6/86.5 | 89.6 | 85.3/89.1 | 70.0 | 91.2 | 87.0/86.8 | 78.36 | 84.12 |
| | | Best (above) | 55.9 | 84.2/83.9 | 90.6/86.5 | 90.7 | 88.3/91.3 | 74.0 | 93.1 | 89.2/88.9 | 83.20 | 86.61 |
| | Three-Stage | 2e-5 | 57.1 | 84.0/84.4 | 90.8/86.5 | 90.9 | 88.1/91.1 | 72.6 | 92.9 | 89.5/89.2 | 83.22 | 86.49 |
| | | 1e-4 | 53.8 | 83.5/84.1 | 90.8/86.8 | 90.1 | 88.3/91.3 | 72.6 | 92.8 | 88.2/87.9 | 82.58 | 86.18 |
| | | 5e-4 | 31.3 | 80.0/81.3 | 89.8/85.3 | 89.2 | 86.7/90.0 | 70.8 | 91.3 | 86.4/86.8 | 78.40 | 84.29 |
| | | Best (above) | 57.1 | 84.0/84.4 | 90.8/86.8 | 90.9 | 88.3/91.3 | 72.6 | 92.9 | 89.5/89.2 | 83.28 | 86.55 |

Table C.3: 2-bit quantization for BERT$_{base}$ with size 26.8MB ($\times 16.0$ smaller than the fp32 version).

| Training Cost | Stages | learning rate | CoLA Mcc | MNLI-m/-mm Acc/Acc | MRPC F1/Acc | QNLI Acc | QQP F1/Acc | RTE Acc | SST-2 Acc | STS-B Pear/Spea | Avg. | Avg. w/o CoLA |
|---|---|---|---|---|---|---|---|---|---|---|---|---|
| Budget-A | One-Stage | 2e-5 | 58.0 | 84.2/84.6 | 89.8/84.8 | 91.5 | 88.4/91.4 | 72.2 | 93.5 | 89.4/89.0 | 83.29 | 86.45 |
| | | 1e-4 | 55.8 | 84.5/85.0 | 90.9/87.0 | 91.5 | 88.4/91.4 | 74.0 | 93.5 | 89.6/89.2 | 83.59 | 87.06 |
| | | 5e-4 | 46.1 | 83.8/84.7 | 89.6/85.0 | 90.9 | 88.0/91.2 | 71.8 | 92.8 | 88.8/88.4 | 81.68 | 86.12 |
| | | Best (above) | 58.0 | 84.5/85.0 | 90.9/87.0 | 91.5 | 88.4/91.4 | 74.0 | 93.5 | 89.6/89.2 | 83.83 | 87.06 |
| | Two-Stage | 2e-5 | 56.6 | 84.3/85.0 | 91.0/87.3 | 91.3 | 88.3/91.3 | 71.5 | 93.5 | 89.6/89.1 | 83.38 | 86.72 |
| | | 1e-4 | 52.3 | 84.2/84.8 | 90.2/85.5 | 91.0 | 87.9/91.1 | 72.2 | 92.8 | 89.4/89.1 | 82.59 | 86.38 |
| | | 5e-4 | 29.0 | 78.1/78.8 | 90.3/85.8 | 90.3 | 86.2/89.9 | 68.2 | 91.7 | 87.6/87.5 | 77.71 | 83.80 |
| | | Best (above) | 56.6 | 84.3/85.0 | 91.0/87.3 | 91.3 | 88.3/91.3 | 72.2 | 93.5 | 89.6/89.1 | 83.46 | 86.81 |
| | Three-Stage | 2e-5 | 58.8 | 84.3/84.7 | 90.4/86.8 | 91.4 | 88.0/91.2 | 72.6 | 93.5 | 89.6/89.1 | 83.66 | 86.76 |
| | | 1e-4 | 55.2 | 84.1/84.9 | 90.6/86.3 | 91.1 | 87.9/91.2 | 74.0 | 93.1 | 89.2/88.9 | 83.23 | 86.74 |
| | | 5e-4 | 33.8 | 83.1/83.8 | 88.5/83.6 | 89.8 | 86.4/90.0 | 47.3 | 91.2 | 84.4/84.0 | 76.28 | 81.65 |
| | | Best (above) | 58.8 | 84.3/84.7 | 90.4/86.8 | 91.4 | 88.0/91.2 | 74.0 | 93.5 | 89.6/89.1 | 83.81 | 86.94 |
| Budget-B | One-Stage | 2e-5 | 58.2 | 84.6/84.7 | 91.3/87.7 | 91.4 | 88.4/91.4 | 73.6 | 93.3 | 89.6/89.2 | 83.83 | 87.04 |
| | | 1e-4 | 56.3 | 84.3/84.9 | 90.5/86.3 | 91.3 | 88.4/91.4 | 75.8 | 93.5 | 89.5/89.1 | 83.70 | 87.12 |
| | | 5e-4 | 47.3 | 84.0/84.3 | 90.1/86.3 | 90.5 | 88.3/91.2 | 72.6 | 92.9 | 88.7/88.2 | 81.98 | 86.31 |
| | | Best (above) | 58.2 | 84.6/84.7 | 91.3/87.7 | 91.4 | 88.4/91.4 | 75.8 | 93.5 | 89.6/89.2 | 84.10 | 87.34 |
| | Two-Stage | 2e-5 | 58.4 | 84.3/84.9 | 90.5/86.3 | 91.4 | 88.4/91.3 | 74.0 | 93.5 | 89.7/89.4 | 83.76 | 86.93 |
| | | 1e-4 | 53.6 | 84.0/84.6 | 91.1/87.3 | 91.0 | 88.0/91.2 | 73.3 | 93.5 | 89.0/88.6 | 83.06 | 86.74 |
| | | 5e-4 | 34.7 | 81.5/82.3 | 90.1/85.5 | 90.5 | 85.5/89.2 | 71.1 | 91.6 | 87.6/87.4 | 79.33 | 84.91 |
| | | Best (above) | 58.4 | 84.3/84.9 | 91.1/87.3 | 91.4 | 88.4/91.3 | 74.0 | 93.5 | 89.7/89.4 | 83.87 | 87.05 |
| | Three-Stage | 2e-5 | 59.6 | 84.5/84.7 | 90.9/86.5 | 91.3 | 88.3/91.3 | 73.3 | 93.3 | 89.7/89.3 | 83.80 | 86.82 |
| | | 1e-4 | 55.9 | 84.6/84.8 | 90.7/86.5 | 91.1 | 88.3/91.4 | 74.4 | 93.3 | 89.3/89.0 | 83.48 | 86.92 |
| | | 5e-4 | 35.3 | 81.4/81.8 | 89.9/85.5 | 90.6 | 86.9/90.2 | 72.9 | 91.6 | 87.8/87.6 | 79.68 | 85.22 |
| | | Best (above) | 59.6 | 84.6/84.8 | 90.9/86.5 | 91.3 | 88.3/91.4 | 74.4 | 93.3 | 89.7/89.3 | 83.96 | 87.00 |
| Budget-C | One-Stage | 2e-5 | 59.9 | 84.7/84.4 | 91.0/86.7 | 91.4 | 88.4/91.4 | 74.0 | 93.5 | 89.6/89.2 | 84.00 | 87.01 |
| | Two-Stage | 2e-5 | 59.9 | 84.9/84.7 | 91.2/87.3 | 91.2 | 88.2/91.3 | 73.3 | 93.5 | 89.7/89.4 | 83.98 | 86.99 |
| | Three-Stage | 2e-5 | 59.6 | 84.7/84.3 | 91.0/87.0 | 91.2 | 88.4/91.3 | 74.4 | 93.7 | 89.6/89.2 | 83.98 | 87.03 |

Table C.4: Results without data augmentation. Details of Table 4 for 1-bit quantization on BERT$_{base}$ with three learning rates.

| Cost | learning rate | CoLA Mcc | MNLI-m/-mm Acc/Acc | MRPC F1/Acc | QNLI Acc | QQP F1/Acc | RTE Acc | SST-2 Acc | STS-B Pear/Spea | Avg. all | Avg. w/o CoLA |
|---|---|---|---|---|---|---|---|---|---|---|---|
| Budget-A One-Stage | 2e-5 | 47.3 | 83.1/83.7 | 87.2/80.1 | 90.8 | 87.4/90.7 | 66.8 | 92.2 | 85.5/85.3 | 80.02 | 84.11 |
| | 1e-4 | 48.9 | 83.7/84.6 | 89.0/83.6 | 91.0 | 88.0/91.1 | 71.5 | 92.5 | 87.6/87.5 | 81.61 | 85.70 |
| | 5e-4 | 38.1 | 83.3/84.1 | 88.4/83.6 | 90.1 | 87.8/90.9 | 69.7 | 92.0 | 85.0/85.0 | 79.64 | 84.84 |
| | Best (above) | 48.9 | 83.7/84.6 | 89.0/83.6 | 91.0 | 88.0/91.1 | 71.5 | 92.5 | 87.6/87.5 | 81.61 | 85.70 |
| Budget-C One-Stage | 2e-5 | 55.0 | 84.1/84.0 | 90.0/85.0 | 90.7 | 88.2/91.2 | 73.3 | 92.9 | 88.2/87.9 | 82.71 | 86.18 |
| | 1e-4 | 52.9 | 84.2/84.4 | 89.2/85.0 | 90.4 | 88.3/91.3 | 72.9 | 92.7 | 87.7/87.7 | 82.39 | 86.08 |
| | 5e-4 | 42.2 | 82.8/82.9 | 88.7/83.6 | 89.8 | 87.7/90.8 | 71.1 | 91.4 | 87.0/86.8 | 80.18 | 84.92 |
| | Best (above) | 55.0 | 84.2/84.4 | 90.0/85.0 | 90.7 | 88.3/91.3 | 73.3 | 92.9 | 88.2/87.9 | 82.78 | 86.25 |

Table C.5: Details of Table 5: Layer reduction with different learning rates (i.e., the 2nd column).

| Model (MB) | Learning rate | CoLA Mcc | MNLI-m/-mm Acc/Acc | MRPC F1/Acc | QNLI Acc | QQP F1/Acc | RTE Acc | SST-2 Acc | STS-B Pear/Spea | Avg. | Avg. w/o CoLA |
|---|---|---|---|---|---|---|---|---|---|---|---|
| BERT-base | n/a | 59.7 | 84.9/85.6 | 90.6/86.3 | 92.1 | 88.6/91.5 | 72.2 | 93.2 | 90.1/89.6 | 83.95 | 86.98 |
| TinyBERT$_6$ | 5e-5 | 53.6 | 84.3/84.3 | 90.4/86.3 | 91.2 | 88.5/91.5 | 67.9 | 93.2 | 89.2/88.8 | 82.39 | 85.99 |
| | 1e-4 | 54.4 | 84.6/84.3 | 90.1/85.8 | 91.5 | 88.4/91.5 | 68.6 | 93.3 | 89.1/89.0 | 82.57 | 86.09 |
| | 5e-4 | 45.1 | 84.1/84.2 | 90.1/85.8 | 91.3 | 88.5/91.5 | 69.7 | 93.2 | 89.1/88.7 | 81.56 | 86.11 |
| MiniLM$_6$ | 5e-5 | 55.1 | 84.4/84.4 | 90.4/86.0 | 91.3 | 88.3/91.4 | 70.8 | 93.0 | 89.4/89.0 | 82.87 | 86.34 |
| | 1e-4 | 55.4 | 84.2/84.5 | 89.5/85.3 | 91.3 | 88.4/91.5 | 70.8 | 93.2 | 89.4/89.0 | 82.84 | 86.28 |
| | 5e-4 | 49.5 | 84.1/83.9 | 90.7/86.5 | 91.4 | 88.5/91.5 | 71.8 | 93.3 | 89.1/88.7 | 82.34 | 86.45 |
| Skip-BERT$_6$ | 5e-5 | 56.9 | 84.5/84.9 | 89.7/84.8 | 91.7 | 88.5/91.5 | 71.8 | 93.5 | 89.8/89.4 | 83.27 | 86.56 |
| | 1e-4 | 56.6 | 84.6/84.7 | 90.4/85.8 | 91.8 | 88.5/91.6 | 72.6 | 93.5 | 89.7/89.4 | 83.43 | 86.79 |
| | 5e-4 | 48.5 | 84.3/84.2 | 90.0/85.5 | 91.5 | 88.6/91.6 | 71.5 | 93.5 | 89.4/88.9 | 82.22 | 86.44 |
| Skip-BERT$_5$ | 5e-5 | 57.9 | 84.3/85.1 | 89.8/84.8 | 91.4 | 88.3/91.4 | 72.2 | 93.3 | 89.2/88.9 | 83.29 | 86.46 |
| | 1e-4 | 56.7 | 84.2/84.8 | 90.1/85.5 | 91.4 | 88.3/91.4 | 72.2 | 93.2 | 89.2/88.8 | 83.18 | 86.49 |
| | 5e-4 | 53.6 | 84.1/84.4 | 90.1/85.5 | 91.4 | 88.5/91.5 | 70.4 | 93.3 | 88.6/88.2 | 82.53 | 86.15 |
| Skip-BERT$_4$ | 5e-5 | 53.3 | 83.0/83.3 | 89.9/85.0 | 90.8 | 88.1/91.3 | 70.0 | 93.2 | 88.8/88.4 | 82.08 | 85.67 |
| | 1e-4 | 52.2 | 83.2/83.4 | 90.0/85.3 | 90.7 | 88.2/91.3 | 70.0 | 93.5 | 88.6/88.3 | 82.02 | 85.75 |
| | 5e-4 | 46.8 | 82.9/83.0 | 89.6/85.0 | 90.4 | 87.8/91.0 | 67.9 | 93.1 | 88.3/87.8 | 80.93 | 85.20 |

Table C.6: Selection for student networks from their teacher networks (fine-tuned BERT$_{base}$). The following results follow the same hyperparamters and the learning rate 5e-5.

| Layer Selection | CoLA Mcc | MNLI-m/-mm Acc/Acc | MRPC F1/Acc | QNLI Acc | QQP F1/Acc | RTE Acc | SST-2 Acc | STS-B Pear/Spea | Avg. all | Avg. w/o CoLA |
|---|---|---|---|---|---|---|---|---|---|---|
| BERT-base fp32 | 59.7 | 84.9/85.6 | 90.6/86.3 | 92.1 | 88.6/91.5 | 72.2 | 93.2 | 90.1/89.6 | 83.95 | 86.98 |
| (1) Skip-6 | 56.9 | 84.5/84.9 | 89.7/84.8 | 91.7 | 88.5/91.5 | 71.8 | 93.5 | 89.8/89.4 | 83.27 | 86.56 |
| (2) Top-6 | 52.0 | 83.9/84.2 | 90.0/85.3 | 91.5 | 88.2/91.3 | 69.3 | 93.0 | 88.2/87.9 | 82.08 | 85.84 |
| (3) Bottom-6 | 51.2 | 83.2/83.6 | 89.5/84.6 | 90.1 | 87.8/90.9 | 67.9 | 92.7 | 89.5/89.1 | 81.50 | 85.29 |

Table C.7: Compare between different stages KD. All learning rate set to be 5e-5.

| # | Model (255.2MB) | #-Stage Same Budget | CoLA Mcc | MNLI-m/-mm Acc/Acc | MRPC F1/Acc | QNLI Acc | QQP F1/Acc | RTE Acc | SST-2 Acc | STS-B Pear/Spea | Avg. all | Avg. w/o CoLA |
|---|---|---|---|---|---|---|---|---|---|---|---|---|
| 1 | TinyBERT$_6$ | One | 53.6 | 84.3/84.3 | 90.4/86.3 | 91.2 | 88.5/91.5 | 67.9 | 93.2 | 89.2/88.8 | 82.39 | 85.99 |
| 2 | | Two | 55.3 | 84.2/84.2 | 89.9/85.3 | 91.1 | 88.4/91.4 | 68.6 | 92.9 | 89.2/88.9 | 82.47 | 85.86 |
| 3 | | Three | 54.2 | 84.2/84.4 | 89.9/85.0 | 91.3 | 88.5/91.5 | 70.8 | 93.1 | 89.2/88.8 | 82.63 | 86.19 |
| 4 | MiniLM$_6$-v2 | One | 55.1 | 84.4/84.4 | 90.4/86.0 | 91.3 | 88.3/91.4 | 70.8 | 93.0 | 89.4/89.0 | 82.87 | 86.34 |
| 5 | | Two | 53.8 | 84.4/84.5 | 90.7/87.3 | 91.2 | 88.3/91.3 | 67.9 | 93.1 | 89.7/89.3 | 82.58 | 86.18 |
| 6 | | Three | 54.0 | 84.4/84.6 | 89.1/85.3 | 91.3 | 88.5/91.5 | 70.4 | 93.0 | 89.6/89.2 | 82.68 | 86.26 |
| 7 | Skip-BERT$_6$ | One | 56.9 | 84.5/84.9 | 89.7/84.8 | 91.7 | 88.5/91.5 | 71.8 | 93.5 | 89.8/89.4 | 83.27 | 86.56 |
| 8 | | Two | 56.0 | 84.1/84.2 | 89.7/84.8 | 91.5 | 88.2/91.3 | 71.8 | 93.3 | 89.4/89.1 | 82.93 | 86.30 |
| 9 | | Three | 55.3 | 84.4/84.6 | 89.7/84.8 | 91.7 | 88.5/91.5 | 72.2 | 93.6 | 89.6/89.2 | 83.08 | 86.55 |

Table C.8: 1-/2-bit quantization of smaller model.

| # | Model #-layer | bit | size (MB) | CoLA Mcc | MNLI-m/-mm Acc/Acc | MRPC F1/Acc | QNLI Acc | QQP F1/Acc | RTE Acc | SST-2 Acc | STS-B Pear/Spea | Avg. all | Acc. drop |
|---|---|---|---|---|---|---|---|---|---|---|---|---|---|
| 1 | TinyBERT$_6$[62] | 32 | 255.2 (×1.6) | 54.1 | 84.5/84.5 | 91.0/87.3 | 91.1 | 88.0/91.1 | 71.8 | 93.0 | 89.8/89.6 | 83.02 | – |
| 2 | | 2 | 16.0 (×26.2) | 53.0 | 83.4/83.8 | 91.5/88.0 | 89.9 | 87.2/90.5 | 71.8 | 93.0 | 86.9/86.5 | 82.26 | -0.76 |
| 3 | XTC-BERT$_6$ SIX | 32 | 255.2 (×1.6) | 56.9 | 84.4/84.8 | 90.1/85.5 | 91.3 | 88.4/91.4 | 72.2 | 93.2 | 90.3/90.0 | 83.33 | – |
| 4 | | 2 | 16.0 (×26.2) | 53.8 | 83.6/84.2 | 90.5/86.3 | 90.6 | 88.2/91.3 | 73.6 | 93.6 | 89.0/88.7 | 82.89 | -0.44 |
| 5 | | 1 | 8.0 (×52.3) | 52.3 | 83.4/83.8 | 90.0/85.3 | 89.4 | 87.9/91.1 | 68.6 | 93.1 | 88.4/88.0 | 81.71 | -1.51 |
| 6 | XTC-BERT$_5$ FIVE | 32 | 228.2 (×1.8) | 56.7 | 84.2/84.8 | 90.1/85.5 | 91.4 | 88.3/91.4 | 72.2 | 93.2 | 89.2/88.8 | 83.18 | – |
| 7 | | 2 | 14.2 (×29.3) | 53.9 | 83.3/84.1 | 90.4/86.0 | 90.4 | 88.2/91.2 | 71.8 | 93.0 | 88.4/88.0 | 82.46 | -0.72 |
| 8 | | 1 | 7.1 (×58.5) | 52.2 | 82.9/83.2 | 89.9/85.0 | 88.5 | 87.6/90.8 | 69.3 | 92.9 | 87.3/87.0 | 81.34 | -1.84 |
| 9 | XTC-BERT$_4$ FOUR | 32 | 201.2 (×2.1) | 52.2 | 83.2/83.4 | 90.0/85.3 | 90.7 | 88.2/91.3 | 70.0 | 93.5 | 88.6/88.3 | 82.02 | – |
| 10 | | 2 | 12.6 (×33.2) | 50.3 | 82.5/83.0 | 90.0/85.3 | 89.2 | 87.8/91.0 | 69.0 | 92.8 | 87.9/87.4 | 81.22 | -0.90 |
| 11 | | 1 | 6.3 (×66.4) | 48.3 | 82.0/82.3 | 89.9/85.5 | 87.7 | 86.9/90.4 | 63.9 | 92.4 | 87.1/86.7 | 79.96 | -2.01 |

Table C.9: **1-bit** quantization for Skip-BERT$_6$.

| # | Cost | LoRa (Y/N) | CoLA Mcc | MNLI-m/-mm Acc/Acc | MRPC F1/Acc | QNLI Acc | QQP F1/Acc | RTE Acc | SST-2 Acc | STS-B Pear/Spea | Avg. | Avg. w/o CoLA |
|---|---|---|---|---|---|---|---|---|---|---|---|---|
| 1 | | N | 47.9 | 82.9/83.2 | 90.5/86.3 | 89.7 | 87.8/90.9 | 65.7 | 92.7 | 88.0/87.6 | 80.81 | 84.92 |
| 2 | Budget-A | Y (DIM=1) | 48.1 | 82.9/83.2 | 89.4/84.3 | 89.6 | 87.6/90.8 | 65.3 | 92.4 | 88.1/87.7 | 80.52 | 84.58 |
| 3 | | Y (DIM=8) | 49.7 | 83.4/83.6 | 89.8/85.0 | 89.9 | 87.5/90.8 | 66.8 | 92.8 | 87.8/87.5 | 81.09 | 85.01 |
| 4 | | N | 50.3 | 83.2/83.7 | 90.9/86.8 | 89.3 | 87.9/91.0 | 67.1 | 92.7 | 88.2/88.0 | 81.37 | 85.25 |
| 5 | Budget-B | Y (DIM=1) | 49.9 | 83.3/83.8 | 90.5/86.0 | 89.3 | 88.0/91.1 | 67.5 | 92.7 | 88.3/87.9 | 81.32 | 85.25 |
| 6 | | Y (DIM=8) | 50.9 | 83.2/83.7 | 90.3/85.8 | 89.4 | 88.1/91.2 | 69.0 | 92.8 | 88.3/88.0 | 81.59 | 85.42 |
| 7 | | N | 51.9 | 83.1/83.6 | 90.5/86.0 | 89.3 | 88.2/91.2 | 69.0 | 93.1 | 88.1/87.7 | 81.70 | 85.42 |
| 8 | Budget-C | Y (DIM=1) | 52.0 | 83.2/83.8 | 89.8/86.0 | 89.4 | 88.0/91.1 | 69.3 | 93.0 | 88.2/88.0 | 81.78 | 85.50 |
| 9 | | Y (DIM=8) | 52.6 | 83.3/83.8 | 89.9/85.5 | 89.2 | 88.2/91.2 | 69.3 | 92.8 | 88.2/87.9 | 81.77 | 85.41 |

Table C.10: **2-bit** quantization for Skip-BERT$_6$.

| # | Cost | LoRa (Y/N) | CoLA Mcc | MNLI-m/-mm Acc/Acc | MRPC F1/Acc | QNLI Acc | QQP F1/Acc | RTE Acc | SST-2 Acc | STS-B Pear/Spea | Avg. | Avg. w/o CoLA |
|---|---|---|---|---|---|---|---|---|---|---|---|---|
| 1 | | N | 51.8 | 83.8/83.7 | 90.2/85.8 | 90.6 | 88.2/91.3 | 70.4 | 93.2 | 88.7/88.3 | 82.14 | 85.94 |
| 2 | Budget-A | Y (DIM=1) | 51.4 | 83.7/84.2 | 90.3/85.8 | 90.5 | 88.2/91.3 | 70.8 | 93.5 | 88.9/88.5 | 82.23 | 86.09 |
| 3 | | Y (DIM=8) | 51.6 | 83.6/84.2 | 90.6/86.3 | 90.7 | 88.2/91.3 | 71.1 | 93.1 | 88.9/88.5 | 82.31 | 86.15 |
| 4 | | N | 52.7 | 83.8/84.5 | 90.0/85.3 | 90.3 | 88.4/91.4 | 72.6 | 93.0 | 89.0/88.6 | 82.51 | 86.24 |
| 5 | Budget-B | Y (DIM=1) | 53.6 | 83.6/83.9 | 90.5/86.0 | 90.2 | 88.4/91.4 | 71.1 | 93.2 | 88.9/88.5 | 82.43 | 86.04 |
| 6 | | Y (DIM=8) | 53.3 | 83.9/84.4 | 90.3/85.8 | 90.2 | 88.3/91.3 | 70.8 | 93.0 | 89.0/88.7 | 82.41 | 86.05 |

Table C.11: **1-bit** quantization for Skip-BERT$_6$.

| Training Cost | LoRa | learning rate | CoLA Mcc | MNLI-m/-mm Acc/Acc | MRPC F1/Acc | QNLI Acc | QQP F1/Acc | RTE Acc | SST-2 Acc | STS-B Pear/Spea | Avg. | Avg. w/o CoLA |
|---|---|---|---|---|---|---|---|---|---|---|---|---|
| Budget-A | N | 5e-5 | 45.6 | 82.6/83.0 | 89.3/84.3 | 89.7 | 87.5/90.8 | 63.5 | 92.4 | 87.6/87.3 | 79.94 | 84.24 |
| | | 1e-4 | 47.9 | 82.9/83.2 | 89.4/84.6 | 89.4 | 87.8/90.9 | 64.3 | 92.4 | 88.0/87.6 | 80.40 | 84.46 |
| | | 5e-4 | 40.6 | 81.9/82.6 | 90.5/86.3 | 88.4 | 87.6/90.8 | 65.7 | 92.7 | 86.9/86.7 | 79.54 | 84.41 |
| | | Best (above) | 47.9 | 82.9/83.2 | 90.5/86.3 | 89.7 | 87.8/90.9 | 65.7 | 92.7 | 88.0/87.6 | 80.81 | 84.92 |
| | Y (DIM=1) | 5e-5 | 46.9 | 82.9/83.2 | 88.9/83.6 | 89.6 | 87.6/90.8 | 63.2 | 92.4 | 87.7/87.4 | 80.03 | 84.18 |
| | | 1e-4 | 48.1 | 82.6/83.5 | 89.4/84.3 | 89.2 | 87.6/90.8 | 65.3 | 92.3 | 88.1/87.7 | 80.47 | 84.51 |
| | | 5e-4 | 39.3 | 82.1/82.8 | 89.2/84.3 | 88.6 | 87.5/90.7 | 65.0 | 92.2 | 87.0/86.8 | 79.11 | 84.09 |
| | | Best (above) | 48.1 | 82.9/83.2 | 89.4/84.3 | 89.6 | 87.6/90.8 | 65.3 | 92.4 | 88.1/87.7 | 80.52 | 84.58 |
| | Y (DIM=8) | 5e-5 | 49.7 | 82.8/83.5 | 88.7/83.1 | 89.9 | 87.5/90.8 | 66.8 | 92.7 | 87.7/87.4 | 80.78 | 84.66 |
| | | 1e-4 | 49.6 | 83.4/83.6 | 89.8/85.0 | 89.4 | 87.6/90.8 | 65.7 | 92.8 | 87.8/87.5 | 80.90 | 84.81 |
| | | 5e-4 | 38.4 | 82.3/83.0 | 89.4/84.1 | 88.9 | 87.6/90.8 | 66.4 | 92.0 | 86.8/86.6 | 79.19 | 84.29 |
| | | Best (above) | 49.7 | 83.4/83.6 | 89.8/85.0 | 89.9 | 87.5/90.8 | 66.8 | 92.8 | 87.8/87.5 | 81.09 | 85.01 |
| Budget-B | N | 5e-5 | 50.3 | 83.1/83.7 | 89.2/84.1 | 89.3 | 87.9/91.0 | 66.8 | 92.7 | 88.2/88.0 | 81.02 | 84.86 |
| | | 1e-4 | 49.7 | 83.2/83.7 | 90.9/86.8 | 88.8 | 87.9/91.0 | 67.1 | 92.7 | 88.1/87.8 | 81.23 | 85.18 |
| | | 5e-4 | 45.0 | 81.9/81.9 | 89.1/84.3 | 88.1 | 87.4/90.7 | 66.1 | 92.5 | 85.9/85.7 | 79.60 | 83.92 |
| | | Best (above) | 50.3 | 83.2/83.7 | 90.9/86.8 | 89.3 | 87.9/91.0 | 67.1 | 92.7 | 88.2/88.0 | 81.37 | 85.25 |
| | Y (DIM=1) | 5e-5 | 49.4 | 83.0/83.6 | 89.7/85.0 | 89.3 | 88.1/91.0 | 67.5 | 92.7 | 88.3/87.9 | 81.09 | 85.05 |
| | | 1e-4 | 49.9 | 83.3/83.8 | 90.5/86.0 | 88.8 | 88.0/91.1 | 67.1 | 92.4 | 88.2/87.9 | 81.18 | 85.09 |
| | | 5e-4 | 41.8 | 81.8/82.5 | 89.7/84.8 | 88.0 | 87.7/90.8 | 63.9 | 92.1 | 86.0/85.9 | 79.08 | 83.74 |
| | | Best (above) | 49.9 | 83.3/83.8 | 90.5/86.0 | 89.3 | 88.0/91.1 | 67.5 | 92.7 | 88.3/87.9 | 81.32 | 85.25 |
| | Y (DIM=8) | 5e-5 | 50.9 | 83.2/83.7 | 90.3/85.8 | 89.4 | 87.9/91.0 | 69.0 | 92.8 | 88.3/88.0 | 81.57 | 85.40 |
| | | 1e-4 | 50.6 | 83.2/83.4 | 89.7/84.8 | 89.1 | 88.1/91.2 | 67.5 | 92.8 | 88.1/87.9 | 81.19 | 85.01 |
| | | 5e-4 | 41.7 | 82.2/82.8 | 89.2/84.1 | 88.6 | 87.7/90.8 | 65.3 | 92.1 | 86.0/85.7 | 79.29 | 83.99 |
| | | Best (above) | 50.9 | 83.2/83.7 | 90.3/85.8 | 89.4 | 88.1/91.2 | 69.0 | 92.8 | 88.3/88.0 | 81.59 | 85.42 |
| Budget-C | N | 5e-5 | 51.9 | 82.9/83.6 | 90.2/85.8 | 89.3 | 88.2/91.2 | 69.0 | 93.1 | 88.1/87.7 | 81.66 | 85.38 |
| | | 1e-4 | 50.3 | 83.1/83.6 | 90.5/86.0 | 88.9 | 87.9/91.0 | 67.9 | 92.7 | 87.6/87.4 | 81.23 | 85.10 |
| | | 5e-4 | 44.4 | 81.7/81.9 | 89.6/84.6 | 88.4 | 87.5/90.7 | 64.6 | 92.9 | 87.1/87.0 | 79.59 | 83.99 |
| | | Best (above) | 51.9 | 83.1/83.6 | 90.5/86.0 | 89.3 | 88.2/91.2 | 69.0 | 93.1 | 88.1/87.7 | 81.70 | 85.42 |
| | Y (DIM=1) | 5e-5 | 52.0 | 83.2/83.8 | 89.8/86.0 | 89.4 | 87.9/91.0 | 69.3 | 93.0 | 88.2/88.0 | 81.77 | 85.49 |
| | | 1e-4 | 48.9 | 83.2/83.5 | 90.4/85.8 | 89.1 | 88.0/91.1 | 68.6 | 92.5 | 87.8/87.5 | 81.17 | 85.20 |
| | | 5e-4 | 42.5 | 81.8/82.3 | 89.5/84.6 | 88.4 | 87.6/90.7 | 65.7 | 92.5 | 87.1/86.8 | 79.51 | 84.14 |
| | | Best (above) | 52.0 | 83.2/83.8 | 89.8/86.0 | 89.4 | 88.0/91.1 | 69.3 | 93.0 | 88.2/88.0 | 81.78 | 85.50 |
| | Y (DIM=8) | 5e-5 | 52.6 | 83.3/83.8 | 89.9/85.5 | 89.2 | 88.1/91.1 | 69.3 | 92.8 | 88.2/87.9 | 81.76 | 85.40 |
| | | 1e-4 | 50.2 | 83.0/83.8 | 89.9/85.5 | 89.0 | 88.2/91.2 | 69.0 | 92.7 | 87.9/87.6 | 81.37 | 85.26 |
| | | 5e-4 | 43.0 | 82.1/82.5 | 90.2/85.5 | 88.7 | 87.5/90.7 | 67.5 | 92.4 | 87.2/86.9 | 79.96 | 84.58 |
| | | Best (above) | 52.6 | 83.3/83.8 | 89.9/85.5 | 89.2 | 88.2/91.2 | 69.3 | 92.8 | 88.2/87.9 | 81.77 | 85.41 |

Table C.12: **2-bit** quantization for Skip-BERT$_6$ with various learning rates.

| Training Cost | LoRa | learning rate | CoLA Mcc | MNLI-m/-mm Acc/Acc | MRPC F1/Acc | QNLI Acc | QQP F1/Acc | RTE Acc | SST-2 Acc | STS-B Pear/Spea | Avg. | Avg. w/o CoLA |
|---|---|---|---|---|---|---|---|---|---|---|---|---|
| Budget-A | N | 5e-5 | 50.8 | 83.8/83.7 | 90.0/85.3 | 90.4 | 88.2/91.3 | 70.4 | 93.2 | 88.6/88.2 | 81.94 | 85.84 |
| | | 1e-4 | 51.8 | 83.6/84.2 | 90.2/85.8 | 90.6 | 88.2/91.3 | 70.4 | 92.9 | 88.7/88.3 | 82.14 | 85.94 |
| | | 5e-4 | 41.4 | 83.2/83.5 | 90.1/85.8 | 89.8 | 87.9/91.0 | 68.2 | 92.5 | 87.8/87.4 | 80.32 | 85.19 |
| | | Best (above) | 51.8 | 83.8/83.7 | 90.2/85.8 | 90.6 | 88.2/91.3 | 70.4 | 93.2 | 88.7/88.3 | 82.14 | 85.94 |
| | Y (DIM=1) | 5e-5 | 51.4 | 83.7/84.2 | 90.3/85.8 | 90.5 | 88.2/91.3 | 70.8 | 93.5 | 88.8/88.4 | 82.22 | 86.07 |
| | | 1e-4 | 51.2 | 83.7/84.0 | 90.3/85.8 | 90.4 | 88.1/91.2 | 70.8 | 92.9 | 88.9/88.5 | 82.10 | 85.96 |
| | | 5e-4 | 43.8 | 83.6/83.5 | 89.6/84.8 | 89.5 | 87.8/91.1 | 69.3 | 92.7 | 87.8/87.5 | 80.68 | 85.29 |
| | | Best (above) | 51.4 | 83.7/84.2 | 90.3/85.8 | 90.5 | 88.2/91.3 | 70.8 | 93.5 | 88.9/88.5 | 82.23 | 86.09 |
| | Y (DIM=8) | 5e-5 | 51.6 | 83.6/84.2 | 90.1/85.3 | 90.7 | 88.2/91.3 | 71.1 | 93.0 | 88.9/88.5 | 82.19 | 86.01 |
| | | 1e-4 | 51.0 | 83.6/84.1 | 90.6/86.3 | 90.3 | 88.3/91.3 | 71.1 | 93.1 | 88.8/88.5 | 82.18 | 86.08 |
| | | 5e-4 | 43.8 | 83.3/83.7 | 89.3/84.1 | 89.8 | 88.1/91.2 | 69.0 | 92.2 | 88.0/87.7 | 80.57 | 85.16 |
| | | Best (above) | 51.6 | 83.6/84.2 | 90.6/86.3 | 90.7 | 88.2/91.3 | 71.1 | 93.1 | 88.9/88.5 | 82.31 | 86.15 |
| Budget-B | N | 5e-5 | 52.5 | 83.5/84.3 | 89.7/84.8 | 90.3 | 88.3/91.3 | 71.5 | 93.0 | 89.0/88.6 | 82.24 | 85.96 |
| | | 1e-4 | 52.7 | 83.8/84.5 | 90.0/85.3 | 90.1 | 88.4/91.4 | 72.6 | 93.0 | 88.8/88.6 | 82.48 | 86.20 |
| | | 5e-4 | 45.2 | 83.3/83.4 | 89.7/85.0 | 89.9 | 88.0/91.1 | 69.0 | 92.7 | 88.3/87.9 | 80.88 | 85.34 |
| | | Best (above) | 52.7 | 83.8/84.5 | 90.0/85.3 | 90.3 | 88.4/91.4 | 72.6 | 93.0 | 89.0/88.6 | 82.51 | 86.24 |
| | Y (DIM=1) | 5e-5 | 52.8 | 83.6/83.9 | 90.0/85.3 | 90.2 | 88.4/91.3 | 71.1 | 93.1 | 88.9/88.5 | 82.24 | 85.92 |
| | | 1e-4 | 53.6 | 83.6/83.8 | 90.5/86.0 | 89.9 | 88.4/91.4 | 71.1 | 93.2 | 88.8/88.4 | 82.38 | 85.98 |
| | | 5e-4 | 46.1 | 83.1/83.9 | 89.7/85.0 | 89.7 | 88.0/91.1 | 68.2 | 92.9 | 88.1/87.9 | 80.90 | 85.25 |
| | | Best (above) | 53.6 | 83.6/83.9 | 90.5/86.0 | 90.2 | 88.4/91.4 | 71.1 | 93.2 | 88.9/88.5 | 82.43 | 86.04 |
| | Y (DIM=8) | 5e-5 | 53.3 | 83.9/84.4 | 89.8/85.0 | 90.2 | 88.3/91.3 | 70.8 | 93.0 | 88.9/88.5 | 82.31 | 85.94 |
| | | 1e-4 | 53.1 | 83.9/84.0 | 90.3/85.8 | 90.1 | 88.4/91.3 | 70.8 | 92.9 | 89.0/88.7 | 82.32 | 85.98 |
| | | 5e-4 | 48.7 | 83.2/83.5 | 89.7/85.5 | 90.0 | 88.3/91.3 | 70.4 | 92.8 | 88.0/87.7 | 81.49 | 85.59 |
| | | Best (above) | 53.3 | 83.9/84.4 | 90.3/85.8 | 90.2 | 88.3/91.3 | 70.8 | 93.0 | 89.0/88.7 | 82.41 | 86.05 |

Table C.13: LoRa and quantization of five-/four-layer model. The learning rate is $2e-5$

| #-layer | #-bit | loRa Y/N | size (MB) | CoLA Mcc | MNLI-m/-mm Acc/Acc | MRPC F1/Acc | QNLI Acc | QQP F1/Acc | RTE Acc | SST-2 Acc | STS-B Pear/Spea | Avg. all | Avg. w/o CoLA |
|---|---|---|---|---|---|---|---|---|---|---|---|---|---|
| Five | 32 | | 228.2 (×1.8) | 56.7 | 84.2/84.8 | 90.1/85.5 | 91.4 | 88.3/91.4 | 72.2 | 93.2 | 89.2/88.8 | 83.18 | 86.49 |
| | 1 | N | 7.1 (×58.5) | 52.2 | 82.9/83.2 | 89.9/85.0 | 88.5 | 87.6/90.8 | 69.3 | 92.9 | 87.3/87.0 | 81.34 | 84.99 |
| | | Y | 7.3 (×57.4) | 51.8 | 81.8/81.9 | 90.2/85.5 | 88.6 | 87.1/90.3 | 68.6 | 92.8 | 87.5/87.2 | 80.98 | 84.62 |
| | 2 | N | 14.2 (×29.3) | 53.9 | 83.3/84.1 | 90.4/86.0 | 90.4 | 88.2/91.2 | 71.8 | 93.0 | 88.4/88.0 | 82.46 | 86.02 |
| | | Y | 14.6 (×28.7) | 54.2 | 83.5/83.9 | 90.4/86.0 | 90.0 | 88.0/91.1 | 72.9 | 93.1 | 88.5/88.1 | 82.58 | 86.12 |
| Four | 32 | | 201.2 (×2.1) | 52.2 | 83.2/83.4 | 90.0/85.3 | 90.7 | 88.2/91.3 | 70.0 | 93.5 | 88.6/88.3 | 82.02 | 85.75 |
| | 1 | N | 6.3 (×66.4) | 48.3 | 82.0/82.3 | 89.9/85.5 | 87.7 | 86.9/90.4 | 63.9 | 92.4 | 87.1/86.7 | 79.96 | 83.91 |
| | | Y | 6.4 (×65.2) | 48.5 | 81.5/80.8 | 89.8/85.3 | 87.6 | 86.9/90.3 | 64.3 | 92.8 | 87.0/86.7 | 79.79 | 83.70 |
| | 2 | N | 12.6 (×33.2) | 50.3 | 82.5/83.0 | 90.0/85.3 | 89.2 | 87.8/91.0 | 69.0 | 92.8 | 87.9/87.4 | 81.22 | 85.09 |
| | | Y | 12.8 (×32.6) | 49.8 | 82.8/82.6 | 90.1/85.5 | 89.1 | 87.8/91.0 | 68.6 | 92.9 | 87.9/87.5 | 81.13 | 85.05 |

Table C.14: More quantization of five-/four-layer model.

| # | Model | size | CoLA Mcc | MNLI-m/-mm Acc/Acc | MRPC F1/Acc | QNLI Acc | QQP F1/Acc | RTE Acc | SST-2 Acc | STS-B Pear/Spea | Avg. | Avg. w/o CoLA |
|---|---|---|---|---|---|---|---|---|---|---|---|---|
| 1 | 1-bit SkipBERT$_5$ | 7.1 (×58.5) | 52.2 | 82.9/83.2 | 89.9/85.0 | 88.5 | 87.6/90.8 | 69.3 | 92.9 | 87.3/87.0 | 81.34 | 84.99 |
| Continue with above using another one-stage |||||||||||||
| 2 | 1-bit | 7.1 (×58.5) | 52.8 | 83.0/83.4 | 89.8/84.8 | 89.6 | 87.8/91.0 | 69.3 | 93.2 | 87.6/87.3 | 81.63 | 85.24 |
| 3 | 1-bit+LoRa1 | 7.3 (×57.4) | 53.0 | 83.0/83.5 | 89.5/84.6 | 89.6 | 87.9/91.0 | 69.3 | 93.2 | 87.6/87.3 | 81.64 | 85.23 |
| 4 | 2-bit SkipBERT$_5$ | 14.2 (×29.3) | 53.9 | 83.3/84.1 | 90.4/86.0 | 90.4 | 88.2/91.2 | 71.8 | 93.0 | 88.4/88.0 | 82.46 | 86.02 |
| Continue with above using another one-stage |||||||||||||
| 5 | 1-bit | 7.1 (×58.5) | 49.7 | 82.4/83.0 | 89.9/85.3 | 88.2 | 87.8/90.9 | 66.8 | 92.7 | 86.4/86.1 | 80.60 | 84.46 |
| 6 | 1-bit+LoRa1 | 7.3 (×57.4) | 48.6 | 82.6/83.1 | 89.9/85.0 | 88.2 | 87.7/90.8 | 66.4 | 92.9 | 86.5/86.2 | 80.46 | 84.44 |
| 7 | 1-bit SkipBERT$_4$ | 6.3 (×66.4) | 48.3 | 82.0/82.3 | 89.9/85.5 | 87.7 | 86.9/90.4 | 63.9 | 92.4 | 87.1/86.7 | 79.96 | 83.91 |
| Continue with above using another one-stage |||||||||||||
| 8 | 1-bit | 6.3 (×66.4) | 49.8 | 82.4/82.3 | 89.5/84.8 | 88.4 | 87.6/90.8 | 63.5 | 92.9 | 87.2/86.9 | 80.23 | 84.04 |
| 9 | 1-bit+LoRa1 | 6.4 (×65.2) | 49.6 | 82.4/82.1 | 89.6/85.0 | 88.5 | 87.5/90.8 | 63.5 | 93.1 | 87.2/86.9 | 80.24 | 84.08 |
| 10 | 2-bit SkipBERT$_4$ | 12.6 (×33.2) | 50.3 | 82.5/83.0 | 90.0/85.3 | 89.2 | 87.8/91.0 | 69.0 | 92.8 | 87.9/87.4 | 81.22 | 85.09 |
| Continue with above using another one-stage |||||||||||||
| 11 | 1-bit | 6.3 (×66.4) | 46.2 | 81.8/82.0 | 89.1/84.6 | 87.2 | 87.3/90.6 | 63.2 | 92.5 | 85.7/85.4 | 79.31 | 83.45 |
| 12 | 1-bit+LoRa1 | 6.4 (×65.2) | 47.9 | 81.4/82.0 | 89.4/85.0 | 87.1 | 87.4/90.6 | 64.3 | 92.9 | 85.7/85.4 | 79.66 | 83.63 |

Table C.15: Compare between teachers: a twelve-layer BERT$_{base}$ vs a six-layer SkipBERT$_6$.

| #-bit | LoRa (Size) Y/N (MB) | Teacher | CoLA Mcc | MNLI-m/-mm Acc/Acc | MRPC F1/Acc | QNLI Acc | QQP F1/Acc | RTE Acc | SST-2 Acc | STS-B Pear/Spea | Avg. | Avg. w/o CoLA |
|---|---|---|---|---|---|---|---|---|---|---|---|---|
| 1-bit | N (8.0MB) | BERT$_{base}$ | 52.3 | 83.4/83.8 | 90.0/85.3 | 89.4 | 87.9/91.1 | 68.6 | 93.1 | 88.4/88.0 | 81.71 | 85.39 |
| | | Skip-BERT$_6$ | 52.0 | 83.2/83.5 | 89.7/85.0 | 89.3 | 87.7/91.0 | 67.5 | 93.2 | 87.9/87.6 | 81.40 | 85.08 |
| | | Difference | +0.3 | +0.2/+0.3 | +0.3/+0.3 | +0.1 | +0.2/+0.1 | +1.1 | -0.1 | +0.5/+0.4 | +0.31 | +0.31 |
| | Y (8.1MB) | BERT$_{base}$ | 52.1 | 83.3/84.0 | 89.9/85.5 | 89.5 | 88.0/91.1 | 68.2 | 93.0 | 88.5/88.1 | 81.69 | 85.39 |
| | | Skip-BERT$_6$ | 52.4 | 83.2/83.5 | 90.4/85.8 | 89.1 | 87.8/91.0 | 68.2 | 93.1 | 87.9/87.6 | 81.58 | 85.22 |
| | | Difference | -0.3 | +0.1/+0.5 | -0.5/-0.3 | +0.1 | +0.2/+0.1 | 0.0 | -0.1 | +0.6/+0.5 | +0.11 | +0.17 |
| 2-bit | N (16.0MB) | BERT$_{base}$ | 53.8 | 83.6/84.2 | 90.5/86.3 | 90.6 | 88.2/91.3 | 73.6 | 93.6 | 89.0/88.7 | 82.89 | 86.53 |
| | | Skip-BERT$_6$ | 54.5 | 83.7/83.8 | 90.5/86.0 | 90.3 | 88.1/91.2 | 71.5 | 93.3 | 88.8/88.4 | 82.57 | 86.07 |
| | | Difference | -0.7 | -0.1/+0.4 | +0.0/+0.3 | +0.3 | +0.1/+0.1 | +2.1 | +0.3 | +0.2/+0.3 | +0.32 | +0.46 |
| | Y (16.2MB) | BERT$_{base}$ | 55.2 | 83.7/84.4 | 90.3/86.0 | 90.5 | 88.2/91.3 | 71.8 | 93.2 | 89.0/88.6 | 82.79 | 86.24 |
| | | Skip-BERT$_6$ | 54.3 | 83.7/83.8 | 90.4/86.0 | 90.4 | 88.0/91.2 | 72.6 | 93.2 | 88.9/88.5 | 82.68 | 86.22 |
| | | Difference | -0.9 | 0.0/+0.6 | -0.1/+0.0 | +0.1 | +0.2/+0.1 | -0.8 | +0.0 | +0.1/+0.1 | +0.11 | +0.02 |

Table C.16: **1-bit** quantization with various random seeds (learning rate here all set to be 2e-5).

| Model Cost | KD #-Stage | Random seed | CoLA Mcc | MNLI-m/-mm Acc/Acc | MRPC F1/Acc | QNLI Acc | QQP F1/Acc | RTE Acc | SST-2 Acc | STS-B Pear/Spea | Avg. |
|---|---|---|---|---|---|---|---|---|---|---|---|
| | fp32 | | 54.2 | 84.25/84.38 | 89.85/85.05 | 91.3 | 88.49/91.45 | 70.8 | 93.1 | 89.18/88.79 | 82.64 |
| TinyBERT$_6$ Budget-B | One | 42 | 50.5 | 82.72/82.95 | 89.08/84.07 | 89.3 | 87.72/90.82 | 65.0 | 92.7 | 87.93/87.64 | 80.66 |
| | | 111 | 49.6 | 82.75/83.15 | 89.04/84.07 | 89.4 | 87.8/90.97 | 65.0 | 92.4 | 87.95/87.75 | 80.59 |
| | | 222 | 48.4 | 82.72/83.08 | 89.11/84.07 | 89.4 | 87.61/90.84 | 64.6 | 92.7 | 88.14/87.75 | 80.43 |
| | | Ave. above | 49.5 | 82.73/83.06 | 89.08/84.07 | 89.4 | 87.71/90.88 | 64.9 | 92.6 | 88.01/87.71 | 80.56 |
| | | std above | 0.9 | 0.01/0.08 | 0.03/0 | 0.0 | 0.08/0.07 | 0.2 | 0.1 | 0.09/0.05 | 0.09 |
| | Two | 42 | 48.8 | 82.58/82.81 | 88.74/83.58 | 89.2 | 87.75/90.87 | 65.0 | 92.4 | 88.2/87.82 | 80.37 |
| | | 111 | 48.9 | 82.84/83.31 | 88.93/83.82 | 89.0 | 87.72/90.87 | 64.3 | 92.7 | 88.4/88.06 | 80.45 |
| | | 222 | 48.6 | 82.72/83.27 | 89.56/84.8 | 88.9 | 87.68/90.89 | 65.7 | 92.6 | 88.2/87.87 | 80.62 |
| | | Ave. above | 48.8 | 82.71/83.13 | 89.08/84.07 | 89.0 | 87.72/90.88 | 65.0 | 92.6 | 88.27/87.92 | 80.48 |
| | | std above | 0.1 | 0.11/0.23 | 0.35/0.53 | 0.1 | 0.03/0.01 | 0.6 | 0.1 | 0.09/0.1 | 0.10 |
| MiniLM$_6$ Budget-B | fp32 | | 54.0 | 84.42/84.58 | 89.13/85.29 | 91.3 | 88.53/91.46 | 70.4 | 93.0 | 89.56/89.19 | 82.67 |
| | Three | 42 | 48.2 | 83.03/83.24 | 88.47/83.33 | 89.2 | 87.85/90.87 | 64.3 | 92.6 | 88.37/88.06 | 80.34 |
| | | 111 | 47.7 | 83.2/83.42 | 88.56/82.84 | 89.4 | 87.76/90.9 | 65.3 | 92.6 | 88.48/88.17 | 80.43 |
| | | 222 | 47.2 | 83.36/83.44 | 89.04/83.82 | 89.4 | 87.88/90.84 | 64.6 | 92.6 | 88.57/88.23 | 80.43 |
| | | Ave. above | 47.7 | 83.2/83.37 | 88.69/83.33 | 89.3 | 87.83/90.91 | 64.7 | 92.6 | 88.47/88.15 | 80.40 |
| | | std above | 0.4 | 0.13/0.09 | 0.25/0.4 | 0.1 | 0.05/0.04 | 0.5 | 0.0 | 0.08/0.07 | 0.04 |
| SkipBERT$_6$ Budget-B | fp32 | | 56.1 | 84.37/84.53 | 90.52/86.03 | 91.6 | 88.34/91.42 | 73.7 | 93.5 | 89.3/88.92 | 83.39 |
| | Three | 42 | 51.1 | 82.79/83.59 | 89.95/85.05 | 89.4 | 88.01/91.07 | 67.5 | 92.9 | 88.09/87.73 | 81.28 |
| | | 111 | 50.7 | 82.96/83.2 | 90.02/85.54 | 89.4 | 87.96/91.04 | 67.9 | 92.9 | 88.35/87.99 | 81.32 |
| | | 222 | 51.8 | 83.02/83.58 | 90.54/86.27 | 89.4 | 87.91/91 | 67.2 | 92.6 | 88.23/87.98 | 81.43 |
| | | Ave. above | 51.2 | 82.92/83.46 | 90.17/85.62 | 89.4 | 87.96/91.04 | 67.5 | 92.8 | 88.22/87.9 | 81.35 |
| | | std above | 0.4 | 0.1/0.18 | 0.26/0.5 | 0.0 | 0.04/0.03 | 0.3 | 0.2 | 0.11/0.12 | 0.06 |

Table C.17: **2-bit** quantization with various random seeds (learning rate here all set to be 2e-5).

| Model Cost | KD #-Stage | Random seed | CoLA Mcc | MNLI-m/-mm Acc/Acc | MRPC F1/Acc | QNLI Acc | QQP F1/Acc | RTE Acc | SST-2 Acc | STS-B Pear/Spea | Avg. |
|---|---|---|---|---|---|---|---|---|---|---|---|
| SkipBERT$_6$ Budget-B | Three | 42 | 53.2 | 83.67/83.79 | 91.25/87.25 | 90.4 | 88.42/91.4 | 72.2 | 93.6 | 88.77/88.36 | 82.70 |
| | | 111 | 53.3 | 83.63/83.59 | 90.91/86.76 | 90.4 | 88.32/91.38 | 72.2 | 93.5 | 88.65/88.32 | 82.60 |
| | | 222 | 54.1 | 83.65/84.27 | 90.82/86.52 | 90.6 | 88.29/91.4 | 71.5 | 93.6 | 88.69/88.38 | 82.70 |
| | | Ave. above | 53.6 | 83.65/83.88 | 90.99/86.84 | 90.4 | 88.34/91.39 | 72.0 | 93.5 | 88.7/88.35 | 82.66 |
| | | std above | 0.4 | 0.02/0.29 | 0.19/0.3 | 0.1 | 0.06/0.01 | 0.3 | 0.1 | 0.05/0.02 | 0.05 |