# OpenReview forum: "XTC: Extreme Compression for Pre-trained Transformers Made Simple and Efficient"
_NeurIPS.cc/2022/Conference — NeurIPS 2022 Accept_

### Official Review · Reviewer_qYxw · 2022-07-10

**Rating:** 7
**Confidence:** 3
**Soundness:** 3 good
**Presentation:** 4 excellent
**Contribution:** 3 good

**Summary:**

The paper provides a thorough understanding of the low-bit (1-bit and 2-bit) quantization for pre-trained BERT. Specifically, the authors discover four characteristics: 1) longer training and proper hyperparameter reduce the gap, 2) 1-stage knowledge distillation is okay, 3) data augmentation is essential, and 4) pre-training distillation for model compression may not be always helpful. From the findings, the paper proposes XtrmC, a model compression technique that integrates the abovementioned points. XtrmC includes a lightweight layer reduction (based on simple rules) followed by long QAT with 1-step KD and DA. The experiments show that the compression ratio of XtrmC is much higher than the previous works, such as TernaryBERT and BinaryBERT.

**Questions:**

Are there any advantages of previous approaches? For example, ternary-to-binary gradual quantization, or 2-step/3-step KD? Maybe the QAT time can be reduced?

Does the longer training (Bucket-C) schedule acceptable computational cost? I basically agree that previous models are under-trained, but would be x2 or x3 epochs too heavy for practical usage?

How about the sensitivity to the random seed? BERT fine-tuning is known to be very sensitive to the random seed [1][2]. Is the proposed XtrmC robust to such randomness?

[1] Fine-Tuning Pretrained Language Models: Weight Initializations, Data Orders, and Early Stopping

[2] On the Stability of Fine-Tuning BERT: Misconceptions, Explanations, and Strong Baselines (ICLR 2021)


**Limitations:**

The authors well addressed the limitations and future research directions.

**Strengths And Weaknesses:**

Overall, the paper is clearly written and well organized. Previous works are sufficiently compared, and the motivation is reasonable. The paper is supported by extensive experiments. Considering that there is no unified comparison of the various methods for low-bit BERT quantization, this paper is very timely. Thank you for your hard work!

Especially, it is interesting that pre-training distillation may not be necessary (Finding 4), which can save huge training costs to obtain the final model. It is also good news that Skip-BERT shows comparable and even better results.

However, the findings are somewhat empirical; It would increase the value of the paper if you could provide some intuitions about why the phenomenon occurs for each finding. For example, for Finding 1, how could longer training help to avoid sharp accuracy drop (or, falling into early local minima)? For Finding 3, why DA is especially important for small tasks such as CoLA and RTE?

---

> ### Author Response · Authors · 2022-08-02
> **Response to Reviewer qYxw**
>
> We thank the reviewer’s highly positive assessment and questions.
>
> **Q1**: Are there any advantages of previous approaches? For example, ternary-to-binary gradual quantization, or 2-step/3-step KD? Maybe the QAT time can be reduced?
>
> **A1**: For 2-stage/3-stage KD, we find that if the training budget is small (Budget-A) and with the learning rate 2e-5, 3-stage KD is indeed better than 1-stage and 2-stage KD (see Table3, line 1,5,9). However, such multi-stage KD results are still no better than single-stage KD with longer training iterations or learning rate search (see Average best). We think the techniques introduced by previous work still carry good value, but their complexity, as shown in the table presented in Reviewer 8VbB Q1 (the 2nd reviewer) or Figure 8 in **Supplementary Material**, may hinder their adoption in practice.
>
> -----
>
> **Q2**: Does the longer training (Bucket-C) schedule acceptable computational cost? I basically agree that previous models are under-trained, but would be x2 or x3 epochs too heavy for practical usage?
>
> **A2**: That’s a great question. There are three aspects to it. First, the training cost on fine-tuning tasks is often not a major bottleneck. For instance, even with Budget-C, the cost of our experiments on GLUE benchmark (for instance, the 12-layer Bert_base on MNLI and QQP) is within 1 day on a single A100 GPU. If we trained on a six-layer BERT, it will be within half day.  We think this time cost is acceptable for most researchers/practitioners. Second, from an agility perspective, although the training is longer, the overall human efforts required in this process to deal with the complex pipeline and tuning are significantly reduced. Third, the training epochs do not always need to increase by 2x or 3x. For example, we did not observe any accuracy gain for QNLI (see Table 2) since its dataset size after data augmentation is already quite large (4.27M sentences).  Therefore, for applications that already have a large amount of data, we do not need to increase the training budget significantly.
>
> ---
>
> **Q3**: How about the sensitivity to the random seed? BERT fine-tuning is known to be very sensitive to the random seed [1][2]. Is the proposed XtrmC robust to such randomness?
>
> **A3**: Thanks for the suggestion and the reference. See Table C.16 and C.17. We find that random seed brings some variation. There is a slight high variance for smaller tasks such as CoLA, RTE, but for the overall average GLUE scores, the one-standard deviation is within 0.1%. As such, XTC is not very sensitive to random seeds. We suspect the less sensitivity in our cases might be partially due to the knowledge distillation techniques (not considered in [1][2]).

---

### Official Review · Reviewer_aquR · 2022-07-11

**Rating:** 6
**Confidence:** 3
**Soundness:** 3 good
**Presentation:** 2 fair
**Contribution:** 2 fair

**Summary:**

This paper provides an empirical study on the compression of BERT via distillation and quantization. By fine-tuning a large amount of compressed BERT models, the authors find that longer training iterations with small learning rate can help quantization, without techniques like multi-state quantization or weights split. Based on a series of empirical observations, the authors propose XtrmC that jointly quantizes and distills an uncompressed BERT, and achieves acceptable performance with 50x model size reduction.


**Questions:**

1. In Table 5, if you train a pre-trained tinyBERT with longer training iterations during fine-tuning, how does its performance compare with the Skip-BERT? For example, an empirical study can be done here: adopting the budget-C to fine-tune a 6-layer TinyBERT initialized by a teacher model. It is expected that the general information learnt from the pre-training is helpful.

2. As to finding 2, one-stage distillation with more training budgets and good setting of learning rates can benefit the final performance of the compressed model. But fine-tuning with a good learning setting may not be easy to find, compared with multi-stage distillation?

3. In Sect. 5, it is mentioned the training is using STE during BP, which is to my surprise as numerous better schemes than STE are out there in the literature (especially in those Binary NN training papers), demonstrating much better performance. An example off my head is https://github.com/lmbxmu/RBNN where there is a much more sophisticated approximation than STE.


**Limitations:**

The paper contains valuable engineering insights, yet its presentation is largely hampered by the careless writing and lack of basic proofreading (e.g. where is the mentioned Fig. 4?), as shown in my editorial comments below.


Editorial:
1. In abstract, resource-constraint should be resource-constrained?
2. P.2, bottom, celebrating should be celebrated.
3. Line 51, accuracy “los”?
4. Line 73, extend should be extent
5. Line 144, the 1-norm argument should be w or w_i?
6. Line 150, bucket-A?
7. The positioning of Tables 1 & 2 should be swapped to be in order.
8. Line 178, add “multi-stage knowledge distillation” after “claim”.
9. Line 184, x2.5 should be 2.5x
10. Number of brackets in (1) doesn’t agree!
11. Line 196, Fig. 4 is referenced in text but it is missing!
12. Line 210, Row should be Rows
13. It is clumsy in having the Best (above) rows in Table 3, why don’t you just bold the winners?
14. P.7, middle paragraph, I am confused by BERT-base and BERT_{base} notations.
15. What is Row 0 in Table 5?
16. Line 280, what is “better a trained”?
17. Line 317, “when binarization”?


**Strengths And Weaknesses:**

Strengths :
1. The experimental results in this paper are abundant. The authors report a series of results under different settings, and present the corresponding findings in a Q&A fashion.

2. The newly found observation helps to build an efficient compressed BERT without a complicated and ad hoc training strategy.

Weaknesses:
1. The novelty of this paper is incremental since it mainly discusses the effectiveness of the strategies proposed in the TinyBERT, TenaryBERT and BinaryBERT. In other words, it is more an engineering or application note, rather than a work of research innovation.

2. Finding 3 is similar to the discussions already provided in TinyBERT and has been verified in its subsequent works.

3. Lack of the discussion on other BERT variants like RoBERTa and generative models like GPT.

---

> ### Author Response · Authors · 2022-08-02
> **Response to Reviewer aquR**
>
> We thank the reviewer’s positive assessment and questions. We are also grateful for reviewer's careful reading and pointing out the typos.
>
> **Q1**: In Table 5, if you train a pre-trained tinyBERT with longer training iterations during fine-tuning, how does its performance compare with the Skip-BERT? For example, an empirical study can be done here: adopting the budget-C to fine-tune a 6-layer TinyBERT initialized by a teacher model. It is expected that the general information learnt from the pre-training is helpful.
>
> **A1**: We are not sure if we understand your question (it would be great if you could explain a bit). Row 4 and Row 5 in Table 5 are the Budget-C fine-tuning of the pretrained TinyBERT/MiniLMv2.  If you are talking about multiple-stage distillation methods proposed in TinyBERT, we have the full details of the results in Appendix Table C.7. Overall, we find the pretrained TinyBERT/MiniLMv2 are no better than the SkipBERT initialized from teacher models.   On the other hand, we agree that the general information learnt from the pre-training is useful, and please note that we are not discarding that information in our compression pipeline. Instead, configurations such as Skip-BERT still reuse partial weights from pre-trained model. What we observe is that, in many cases such information is already sufficient for task-specific compression to achieve competitive accuracy.
>
>
> ---
> **Q2**: As to finding 2, one-stage distillation with more training budgets and good setting of learning rates can benefit the final performance of the compressed model. But fine-tuning with a good learning setting may not be easy to find, compared with multi-stage distillation?
>
> **A2**: We think it is the other way around. Compared with multi-stage distillation, One-stage KD fine-tuning with a good learning setting is simple one. For each independent experiment, recall that there will be *two*/*three* predefined learning rates for Two/Three-stage KD. Thus, in practice, one would need to tune these *two*/*three* learning rates for Two/Three-stage KD instead of one learning rate (using One-stage KD).
>
> In the original “multi-stage distillation" work [TinyBERT, 20], the authors proposed to use a fixed learning rate scaling ratio to decide the learning rate of the subsequent stage (e.g., the second stage has a 2.5x smaller learning rate than the first stage). However, it is unclear how this 2.5x ratio is determined, which we conjecture would require additional tuning effort for the users to figure out.
>
>  ---
>
> **Q3**: In Sect. 5, it is mentioned the training is using STE during BP, which is to my surprise as numerous better schemes than STE are out there in the literature (especially in those Binary NN training papers), demonstrating much better performance. An example off my head is https://github.com/lmbxmu/RBNN where there is a much more sophisticated approximation than STE.
>
> **A3**: Thanks for the suggestion.  As you know, this paper is about empirical investigation on existing methods such as TinyBERT [20], miniLMv2[49] and TernaryBERT[56] and BinaryBERT[3]. We are trying to be fair on the algorithm parts, so we did not change STE to other schemes. We agree that combining more sophisticated gradient approximation schemes with extreme compression is an interesting study, and we will add your comments and leave its exploration as future work.
>
>
> ---
> **Q4**: The paper contains valuable engineering insights, yet its presentation is largely hampered by the careless writing and lack of basic proofreading (e.g., where is the mentioned Fig. 4?), as shown in my editorial comments below.
>
> **A4**: We highly appreciate for reading our paper carefully. We have done multiple careful editing. As for Fig 4., it’s given in appendix (page 16) due to space limit (we will try to move this figure to the main text if space allowed). We corrected everything and uploaded a new version in “**Supplementary Material**” (not the main pdf, we tried not to overlap the previous version) and hope you feel it is clear. Here we answered some:
>
> 7: “The positioning of Tables 1 & 2 should be swapped.”  We have thought about this. We tend to keep this order in that Table 1 is about emphasizing on “convergence with training longer” as explained Figure 3, which is a more straight-forward concept than that of explaining “knowledge distillation”.
>
> 13. “It is clumsy to have the Best (above) rows in Table 3, why don’t you just bold the winners?” Thanks, we removed these rows in Table 3.
>
> 15. “What is Row 0 in Table 5?” It is a typo and we meant Row 1

---

> > ### Comment · Reviewer_aquR · 2022-08-08
> > **Thanks for the response**
> >
> > I think the response has addressed my queries well.

---

### Official Review · Reviewer_8VbB · 2022-07-11

**Rating:** 6
**Confidence:** 4
**Soundness:** 3 good
**Presentation:** 3 good
**Contribution:** 3 good

**Summary:**

The paper presents an empirical study of the role of various stages in recent extreme compression of Transformer architectures for NLP tasks. Based on this study, the paper presents a simplified architecture and training process that significantly reduces model size.

**Questions:**

Why do you think extra training time is effective for quantizing BERT models? Could you speculate as to how this extra training helps?

What would happen if you took the competing methods and merely increased their training times (without changing their architectures)?
Would a finer grid-search on learning rates be effective? Perhaps there is more gain to be had, since training of quantized networks would have rather coarse gradients, and therefore more sensitive to learning rate.

While your method avoids the time penalty of more complex distillation, it requires longer training times. What is the exact tradeoff in computational load of training between your method and the competing methods?

**Limitations:**

The authors do mention some limitations. They do not discuss any potential negative societal impacts, although they could perhaps mention positive social impacts arising from greatly reduced inference time computational loads, as well as reduced training times, both of which would lower the carbon footprint of neural networks.

**Strengths And Weaknesses:**

Provides a thorough empirical study of how the # of training iterations affects heavily quantized network compression techniques. This shows that the accuracy of binarized networks is improved merely by training longer combined with data augmentation.This study also shows that complicated multi-stage distillation is not required to maintain accuracy of heavily quantized networks.
Based on the empirical study, the paper presents a simplified model which consists of layer reduction (structured pruning) and a single stage of distillation. The experimental results show that this simplified model works well compared to the more complex baselines.

As the proposed method requires more training iterations, it may be more computationally complex than competing methods, even if those methods require more complex distillation stages. The paper lacks a detailed analyses of the computational complexity of the various methods.

The contribution of the paper, while effective, is ad-hoc. It would be nice to have some theoretical discussion of "why" increased training times helps with heavy quantization, and why additional distillation stages are not needed. At least the weight distribution histogram could be presented as a way of gauging the effect of training in the low- versus high-training time regimes.

Another weakness is that much of the details of the study is included in the supplementary material, and not in the main text. A neurips paper should be self contained, without requiring reference to supplementary material to gain important details.

---

> ### Author Response · Authors · 2022-08-02
> **Response to  Reviewer 8VbB (part I)**
>
> We thank the reviewer’s positive assessment and questions.
>
> **Q1**: As the proposed method requires more training iterations, it may be more computationally complex than competing methods, even if those methods require more complex distillation stages. The paper lacks a detailed analyses of the computational complexity of the various methods.
>
> **A1**: Thanks for the question. As the proposed method consists of two steps (layer reduction and the quantized layer-reduced method), we gave the analysis in the following tables (we take the mnli task as an example, the (estimated) GPU hours is on NVIDIA A100).
>
> At first glance, our 1-stage distillation may appear to be more computationally heavy than multi-stage distillation because it trains with more iterations. However, note that there are additional optimization stages introduced by existing works before multi-stage distillation in order to perform extreme compression, such as training a ternarized model and pre-training distillation. Those optimizations add non-negligible computation overhead and additional tuning efforts. If we holistically consider the computation complexity of all those optimizations, our proposed method achieves significantly lower training cost and complexity than existing method.  We will add a clarification in the final version (see our Figure 7 at Page18 in the appendix)
>
>
>
> | steps     |                                 Step I: Layer reduction (TinyBERT[20])  |                                                                                                                      |                                 Step II: 1-bit quantization (BinaryBERT [3])        |                                                                      |                                           |                                                                                                                      |
> |-----------|-------------------------------------------------------------------------|----------------------------------------------------------------------------------------------------------------------|-------------------------------------------------------------------------------------|----------------------------------------------------------------------|-------------------------------------------|----------------------------------------------------------------------------------------------------------------------|
> | methods   |                         General KD pretraining                          |                         KD stage-I  and KD stage-II                                                                  |                         dynaBERT                                                    |                         TernaryBert Training                         |                         Weight splitting  |                      KD   stage-I  and KD stage-II                                                                   |
> | gpu hours |                   ~300   GPU hours on training large-scale text corpus  |                         ~24 GPU hours: Tuning Efforts on learning rate and training iterations for   various stages  |                         ~24 GPU hours  to train half-width model                    |                         MNLI: ~12 GPU hours to train ternary weight  |                                           |                         ~12 GPU hours: Tuning Efforts on learning rate and training iterations   for various stages  |
>
>
> | steps     |                              Step   I: Layer reduction                 |                              Step   II: 1-bit quantization        |
> |-----------|------------------------------------------------------------------------|-------------------------------------------------------------------|
> | ours      |                         (Lightweight layer reduction) Single-stage KD  |                         Single-stage KD                           |
> | GPU hours | ~24 GPU hours                                                          | ~24 GPU hours                                                     |

---

> > ### Author Response · Authors · 2022-08-02
> > **Response to Reviewer 8VbB (part II)**
> >
> > Q2: It would be nice to have some theoretical discussion of "why" increased training times helps with heavy quantization, and why additional distillation stages are not needed.
> >
> > **A2**: Thanks for the suggestion. Based on your comment, we added Figure 5 (Page 17) and Figure 6 (Page 18) in Appendix C to the updated version.
> >
> > Figure 5 is for understanding “why increased training times helps with heavy quantization”. We compare the 1-bit quantized weights between Budget-A and Budget-C. The top plot is to measure the L2-norm distance between the 1-bit SkipBERT_6 and its fp32 counterpart (the initialization), and the bottom plot is to understand the distributions of quantized weight matrices. We see (the top plot) that comparing with Budget-A, Budget-C training results in a longer distance between its initialization and the best quantized models. In addition, the bottom plot shows that many values concentrate closer to 0.045 or –0.045, while the peak frequency under Budget–A concentrates closer to 0.03 or –0.04 (the absolute values are smaller than that of Budget-C). These results indicate that training longer indeed helps the binarized model to make more updates to the weights, which presumably leads to a better solution.
> >
> > Figure 6 is for understanding “why additional distillation stages are not needed”. Like Figure 5, we plot the L2-distance between the 1-bit BERT_base and its fp32 counterpart (the teacher model), and the distributions of quantized weight matrices. We see that the green and red curves overlap, and the distributions are quite similar, which means one-stage and multi-stage could result in similar solutions.
> >
> >
> > ---
> > Q3: Another weakness is that much of the details of the study is included in the supplementary material, and not in the main text. A neurips paper should be self contained, without requiring reference to supplementary material to gain important details.
> >
> > **A3**: Thank you for pointing this out. As you can see from the Appendix, we have done quite a comprehensive evaluation of many different optimizations. Given the page limit of NeurIPS, we tried to focus a bit more on presenting the key findings and our proposed method in the main paper. Based on this feedback, we will try to cut some of the results in the main text (such as the results for data augmentation) and bring the analysis from Appendix back to main text. If the paper gets accepted, we will also use the extra page for presenting more analysis in the main text.
> >
> > ---
> > Q4: Why do you think extra training time is effective for quantizing BERT models? Could you speculate as to how this extra training helps?
> >
> > A4:  As we know BinaryBert only has –1 and 1 value with a scaling factor, so it will take a long time to converge (show in Figure 3).  In addition, using a large learning rate will probably make the training unstable or even diverge, because it will cause the signs of the quantized weights frequently flip. Thus a small learning rate makes sure the model learns appropriately and long training time makes sure the model converges.
> >
> > ---
> > Q5: What would happen if you took the competing methods and merely increased their training times (without changing their architectures)? Would a finer grid-search on learning rates be effective? Perhaps there is more gain to be had, since training of quantized networks would have rather coarse gradients, and therefore more sensitive to learning rate.
> >
> >
> > A5: If you are referring to “multiple-stage distillation” used in existing works such as [TinyBERT, BinaryBERT] as the “competing methods”, we can answer that “multiple-stage distillation” is probably not as effective as single-stage KD if learning rate search is used.  You may take a look at Appendix Table C.1 where we include fine-tuning with multiple stages of KD with a longer training time under three learning rate searches. As shown, longer training iterations help improve the performance of multi-stage KD as well if comparing it with its short training counterpart. However, under the same training budget-C, the multi-stage KD performance is on-par or even slightly worse than one-stage KD. In addition, we performed some comparisons on fine-tuning (and quantization) TinyBert and MiniLM with multiple stage distillation. The fine-tuning results are given in Table C.7 (Training Budget-C). The quantization results are given in Appendix Table C.16.  As can be seen, multiple-stage KD does not provide better results than 1-stage KD under the same training budget. It is possible that a finer grid-search of learning rates of each stage might provide better results. However, that would come as a cost of increased tuning efforts.
> >
> > ---
> > Q6: While your method avoids the time penalty of more complex distillation, it requires longer training times. What is the exact tradeoff in computational load of training between your method and the competing methods?
> >
> > **A6**: Please see our answer to Q1. Let us know if it does not answer your question.

---

> > > ### Comment · Reviewer_8VbB · 2022-08-09
> > > **"why increased training time helps",.**
> > >
> > > Thank you for your answers. However, for the question of "why increased training time helps", I was looking for some theoretical reasoning or even speculation. The answer you gave was just more empirical support for the claim. If one knew the reason for this effect, it could potentially be applied to other problems, and perhaps lead to better learning rate scheduling in general for compressed networks.

---

### Official Review · Reviewer_KREQ · 2022-07-12

**Rating:** 6
**Confidence:** 3
**Soundness:** 3 good
**Presentation:** 4 excellent
**Contribution:** 3 good

**Summary:**

In this work, authors start from a point where they perform a systematic study to measure the impact of many key hyperparameters and training strategies from previous work, such as distillation, DA, etc. Then, they find out that previous baselines for ultra-low bit precision quantization are significantly under-trained. Based on the study, they propose a new compression pipeline named XtrmC. The focus of this paper is mainly on BERT.

**Questions:**

As stated in the weaknesses

**Ethics Review Area:**

["I don’t know"]

**Limitations:**

Yes

**Strengths And Weaknesses:**

Strengths:
- informative and deep study on the existing work. The key factors evaluated by the paper may inspire following research
- Good presentation of the experiment results
- Clear statements on the limitation of the current work and show how the future could be.

Weaknesses:
- The overall discussion in the main draft focus on the study, but not the new method. More dicussion on the new compression method may benefit the overall presentation logic since the title is about the extreme compression, not just a study on existing methods.

---

> ### Author Response · Authors · 2022-08-02
> **Response to Reviewer KREQ**
>
> We thank the reviewer’s positive assessment and questions.
>
> **Q1**: The overall discussion in the main draft focus on the study, but not the new method. More discussion on the new compression method may benefit the overall presentation logic since the title is about the extreme compression, not just a study on existing methods.
>
> **A1**: Thanks for the suggestion. We have a long discussion internally. However, we still tend to think the current format has some advantage of highlighting our *extensive* empirical investigation of the current existing methods. The proposed method is a result of the investigation. In this logic, we would like the readers to focus on both our investigation results and the proposed method.

---

### Meta-Review · Area_Chair_ap9T · 2022-08-27

**Recommendation:** Accept
**Confidence:** Certain

**Metareview:**

Reviewers agree that this paper presents a systematic study on the impact of hyper-parameters and training strategies of previous works. Based on those empirical observations, they propose a simplified model with layer reduction and single-stage distillation, which do not rely on a complicated and ad-hoc training strategy. Extensive experiments are conducted with thorough comparison with existing works. Authors also clearly point-out their current limitations.

The major concern is that this paper is more focused on discussion of the effectiveness of training strategies in previous methods, while the theoretical contribution is somehow limited. It would be much better if authors could explain their observations (more training epochs is needed while additional distillation stages can be discarded) from a theoretical perspective, although this may be far out of the scope of this work. Nonetheless, this paper presents valuable empirical study over existing Transformer compression methods and may inspire following research; therefore, AC recommends acceptance.


**Award:**

No

---

### Decision · Program_Chairs · 2022-09-14

Accept